# Therapeutic Efficacy of CD34-Derived Allogeneic Dendritic Cells Engineered to Express CD93, CD40L, and CXCL13 in Humanized Mouse Models of Pancreatic Cancer

**DOI:** 10.3390/vaccines13070749

**Published:** 2025-07-12

**Authors:** Sara Huerta-Yepez, Jose D. Gonzalez, Neha Sheik, Senay Beraki, Elango Kathirvel, Ariel Rodriguez-Frandsen, Po-Chun Chen, Tiran Sargsyan, Saleemulla Mahammad, Mark R. Dybul, Lu Chen, Francois Binette, Anahid Jewett

**Affiliations:** 1Division of Oral Biology and Medicine, The Jane and Jerry Weintraub Center for Reconstructive Biotechnology, School of Dentistry, University of California, 10833 Le Conte Ave, Los Angeles, CA 90095, USAajewett@mednet.ucla.edu (A.J.); 2Oncology Research Unit, Hospital Infantil de Mexico, Federico Gomez, Mexico City, Mexico 06720, Mexico; 3Renovaro Biosciences, 2080 Century Park East, Suite 906, Los Angeles, CA 90067, USA; jgonzalez@renovarobio.com (J.D.G.); nsheik@renovarobio.com (N.S.); sberaki@renovarobio.com (S.B.); ekathirvel@renovarobio.com (E.K.); smahammad@renovarobio.com (S.M.); lchen@renovarobio.com (L.C.);; 4The Center for Global Health Practice and Impact, Georgetown University Medical Center, 500 First Street, NW, Washington, DC 20002, USA; 5The Jonsson Comprehensive Cancer Center, UCLA School of Dentistry and Medicine, Los Angeles, CA 90095, USA

**Keywords:** CD34 hematopoietic stem cells, dendritic cell therapy, CD40L, CD93, CXCL13, pancreatic cancer, humanized mouse model

## Abstract

**Background/Objectives**: Pancreatic cancer remains the fourth leading cause of cancer-related deaths. While peripheral blood-derived mature dendritic cell (mDC) vaccines have shown potential in eliciting anti-tumor immune responses, clinical efficacy has been limited. This study aimed to enhance the potency and scalability of DC-based immunotherapy by developing an allogeneic DC platform derived from CD34^+^ hematopoietic stem cells (HSCs), genetically engineered to overexpress CD93, CD40L, and CXCL13, followed by maturation and tumor antigen pulsing. **Methods**: Engineered DCs were generated from CD34^+^ HSCs and matured in vitro after lentiviral transduction of CD93, CD40L, and CXCL13. Tumor lysates were used for antigen pulsing. A scrambled-sequence control DC was used for comparison. In vitro assays were performed to assess T cell activation and tumor cell killing. In vivo efficacy was evaluated using orthotopic pancreatic tumors in BLT and PBMC-humanized NSG mice established with the MiaPaca-2 (MP2) cell line. **Results**: Engineered DCs significantly enhanced T cell activation and tumor-specific cytotoxicity in vitro compared to control DCs. Antigen pulsing further amplified immune activation. In vivo, treated humanized mice showed increased CD4^+^, CD8^+^, and NK cell frequencies in peripheral blood and within tumors, correlating with reduced tumor burden. **Conclusions**: Our data shows that the antigen-pulsed, engineered DCs have the potency to activate immune cells, which leads to a significant reduction in pancreatic tumors and therefore could potentially provide an effective therapeutic opportunity for the treatment of pancreatic cancer and other solid tumors.

## 1. Introduction

Pancreatic cancer (PC) is one of the leading causes of cancer-related deaths, and its incidence is rising [1,2]. In the United States, PC remains a major public health concern, with approximately 60,430 new cases and 48,220 deaths reported in 2021, making it the third leading cause of cancer-related deaths after lung and colorectal cancers. The prognosis for PC is poor, with a 5-year survival rate of less than 10% [2]. Only 10–20% of patients with locally advanced or metastatic disease are eligible for curative surgeries [3,4], and even after surgical resection, the malignancy has a high recurrence rate requiring further treatments such as chemotherapy, radiation, or targeted drug therapies [5]. Over the past few decades, both incidence and mortality have steadily increased in the U.S., underscoring the urgent need for improved diagnostics, treatments, and preventive strategies. The relationship between the immune system and PC is influenced by multiple factors, including histological variety, tumor size, lymph nodes, and the composition of the tumor microenvironment (TME) [6,7]. A crucial aspect is a patients’ weak immune system, and/or a state of local suppression, which may prevail, conferring resistance to tumor cell-killing by the immune system. As such, this favors cancer progression [8], making PC notoriously resistant to immunotherapy [9,10]. Failure of these existing therapies has been attributed to the scarcity of CD8+ T cells and profound immunosuppression in pancreatic TME [11,12]. Therefore, novel treatment modalities that can overcome tumor immunosuppression are much needed for the treatment of PC.

Dendritic cells (DCs), the primary antigen-presenting cells (APCs), are potent activators of both the innate and adaptive immune system and can successfully induce anti-tumor immunity [13]. DC paucity in PC leads to dysfunctional immune surveillance, and it has been shown that restoring DC presence in early pancreatic lesions reinvigorates anti-tumor T cell immunity [9]. While it is true that there is a growing interest in reprogrammable immune cell types, like macrophages and Natural Killer (NK) cells, DCs are of particular interest due to their superior ability to prime naïve T cells and initiate robust, antigen-specific adaptive immune responses. Unlike macrophages, which primarily act as phagocytes and can exhibit immunosuppressive phenotypes in the TME, DCs are uniquely effective at cross-presentation and driving cytotoxic T cell responses, a key for sustained antitumor immunity. Several DC-vaccination strategies have previously demonstrated clinical and immunological responses in phase I/II clinical trials for cancer therapies [14,15]. According to Shindo et al., MUC1-loaded DC vaccines were shown to be safe and effective when combined with MUC1-loaded cytotoxic T lymphocytes (CTLs) and gemcitabine [15]. Another study demonstrated that the combination of autologous tumor lysate-loaded DC vaccines with temozolomide improved the survival of patients diagnosed with glioblastoma [16]. The above studies confirm that utilizing DCs’ antigen-presenting abilities and heterogeneity offers a therapeutic modality, with significant promise for enhancing the anticancer immune response [17]. The sourcing of DCs is of critical significance, as differentiation pathways and the fate of DCs may play vital roles in antigen presentation efficiency, cross-presentation capacity, cytokine secretion potential, and overall function [17].

In this study, we have chosen to produce DCs from CD34+ human hematopoietic stem cells (HSCs) isolated from a healthy human donor’s mobilized leukopak, based on several advantages over the traditional monocyte-derived (Mo-)DCs. HSCs allow early genetic modifications by lentiviral transduction without compromising cell viability or growth and offer superior expansion capacity and post-thaw cell recovery [18,19]. Although CD34+ derived DCs require an additional differentiation step, the cost of the process is justified by long-term cost-efficiency. In addition, it has been shown in previous studies that DCs derived from HSCs stimulate a greater mixed lymphocyte reaction (MLR) in allogeneic T lymphocytes over Mo-DCs [20]. Furthermore, using an optimized cytokine differentiation process on these DCs can be tailored to express desired immunostimulatory molecules that illicit specific immune responses while maintaining cGMP compliance for clinical use [17]. To amplify the functionality of CD34-derived DCs, we engineered them for cancer therapy by overexpressing CD93, CD40L, and CXCL13. These genes were selected based on their complementary and potential synergistic roles in the modulation of key immunological processes such as antigen presentation, immune cell recruitment, optimal T cell engagement, and priming.

CD93 (C1qRp) is a C-type lectin-like type I transmembrane protein that plays a role in the phagocytosis of apoptotic cells and the regulation of innate immunity and inflammation [21]. CD93 is known to be expressed in myeloid lineage cells, platelets, monocytes, microglia, and endothelial cells and is responsible for regulating vascular maturation [22,23]. The function of CD93 is recognized in immune cell infiltration of tumors and enhancement of monocyte phagocytosis, suggesting a crucial role for CD93 in enhancing DC-mediated immune responses, likely by improving antigen uptake and facilitating immune cell recruitment to tumor sites, thereby amplifying T-cell activation [21,24].

CD40 ligand (CD154 and CD40L), also known as TRAP or T-BAM, is a 34–39 kDa type II integral membrane protein that binds to CD40 [25]. It is primarily found in activated CD4+ T cells as well as activated B cells, platelets, and smooth muscle cells [26]. Recent studies have shown functional CD40L expression on various DC subtypes and revealed its role in activating T cells [27,28]. The bidirectional crosstalk between DCs and lymphocytes mediated by CD40–CD40L interactions is critical for immune responses. In T follicular helper cells, CD4OL promotes B cell maturation and function by engaging CD40 on the B cell surface and therefore facilitating cell–cell communication [29]. Stable CD40L expression on DCs enables production of IL-12 to overcome immunosuppression and to trigger memory T cell differentiation and antigen-specific T cell responses [30].

CXCL13, a potent B-cell chemoattractant consistently present in lymphoid tissue, serves as a potential candidate for attracting B cells to DCs [31]. Studies on CXCR5−/− (CXCL13 receptor) mice highlighted the significance of CXCL13 in lymphocyte homing to secondary lymphoid organs, as evidenced by the impaired development of B cell follicles and the absence of certain lymph nodes [32]. Studies have demonstrated that various subtypes of DCs express CXCL13, underscoring the crucial role of CXCL13 in mediating the chemoattraction of B cells and potentially activating antigen-specific T helper cells [33].

We reasoned that DCs can be engineered by expressing CD93, CD40L, and CXCL13 and be optimized to become efficient APCs and create a microenvironment to maximize T cell activation. To assess these immunological responses, in vitro and in vivo models were employed for exploration.

Humanized (Hu-) mouse models are key for studying tumor biology and testing therapeutics with a functional human immune system. The Hu-PBMC (peripheral blood mononuclear cells) model, engrafting immunocompromised mice with human leukocytes from healthy donor blood, spleen, or lymph nodes, offers rapid, cost-effective engraftment, ideal for high-effector and memory T cell studies [34,35]. In this study, Hu-PBMC mice bearing orthotopic pancreatic tumors were used to evaluate the therapeutic efficacy of our allogeneic DC treatment platform utilizing a lentiviral vector carrying CD93, CD40L, and CXCL13 with tumor-specific antigen pulsing. Our data demonstrates that engineered DCs play a crucial role in priming and expanding tumor-specific CD8+ T cells and lead to pancreatic tumor growth inhibition (therapeutic efficacy).

## 2. Materials and Methods

### 2.1. Cells, Reagents, and Antibodies

Cells were cultured in a 5% CO_2_-containing humidified incubator maintained at 37 °C and were evaluated for mycoplasma contamination before use. MP2 cells were purchased from the American Type Culture Collection (ATCC) (CRL-1420) (Manassas, VA, USA) and were cultured as per the manufacturer’s recommendations in Dulbecco’s Modified Essential Medium (DMEM) (Gibco, Waltham, MA, USA) supplemented with fetal bovine serum (FBS) to a final concentration of 10%. Human PBMCs (ATCC [PCS-800-011]), T cells (ATCC [PCS-800-016, PCS-800-017]), SNU 423 cells (ATCC [CRL-5971]) and other immune cells were cultured as per the manufacturer’s recommendations in RPMI 1610 Medium (Gibco) supplemented with FBS to a final concentration of 10%. Alpha-MEM (Modified Essential Medium) (Life Technologies, Carlsbad, CA, USA) was used to culture osteosarcoma cells (OSCSCs), which were isolated from freshly resected tongue tumors obtained from the UCLA lab. A GFP-encoded lentivirus (Addgene, Watertown, MA, USA, Cat#93194) was used to transduce SNU 423 or MP2 cells for use in in vitro assays as per the manufacturer’s guidelines.

Human CD34+ HSCs (Cat: mLP, RegH, CR, CD34+, PS, 10 M Lot: 3091646, AllCells, Alameda, CA, USA) were cultured in CellGenix^®^ GMP SCGM (Stem Cell Growth Medium), supplemented with 10% StemSpan™ CD34+ Expansion Supplement (StemCell Technologies, Vancouver, BC, Canada). Purified CD34+ HSCs were acquired from a commercial vendor, and IRB/REB ethical committee approval and review have been obtained by the vendor (AllCells, Alameda, CA, USA). IMDM (Iscove’s Modified Dulbecco’s Medium) (Irvine Scientific, Irvine, CA, USA) medium containing 5% Human Serum Type AB (Access Biologicals-Grifols, Spain), 35 U/mL rh-SCF, 4000 IU/mL rh-MC-SF, 70 IU/mL rh-IL-3, and 42 U/mL rh-Flt3L from Miltenyi Biotec, Germany, was used during monocyte differentiation of CD34+ HSCs. AIM V medium (Gibco-Thermo Scientific, Waltham, MA, USA) containing 2% Human Serum Type AB, 500 IU/mL rh-GMC-SF and 500 IU/mL rh-IL-4 (Miltenyi) was used during differentiation of monocytes to immature DCs (imDCs).

AIM V medium (Gibco) containing 2% Human Serum Type AB, 500 IU/mL rh-GMC-SF, 500 IU/mL rh-IL-4, 1400 IU/mL rh-IL-6, 1700 IU/mL rh-IL1-b, 600 U/mL rh-IL-12, 3800 U/mL rh-IL-15 (Miltenyi), and 2 µg/mL PG-E2, (Sigma Aldrich, St. Louis, MO, USA) was used during differentiation of imDCs to mDCs and during antigen pulsing, in which lysate was generated from either MP2 cells or from corresponding tumor tissue.

Human M-CSF, mAbs (monoclonal antibodies) for NK cells (anti-CD16) and ELISA kits for IFN-γ were obtained from Biolegend (San Diego, CA, USA). RANKL was obtained from PeproTech (Cranbury, NJ, USA) and rh-IL2 was obtained from Hoffman La Roche (Little Falls, NJ, USA).

Monoclonal antibodies for T cells (human anti-CD3/CD28 mAbs) were purchased from STEMCELL Technologies. REAfinity recombinant mAbs VioBlue (VB)-conjugated anti-CD34 (clone REA1164), fluorescein isothiocyanate (FITC)-conjugated anti-CD93 (clone REA1111), allophycocyanin-Vio 770 (APC-Vio 770)-conjugated anti-CD40L (clone REA238), PE (phycoerythrin) and APC-conjugated anti-CD14 (clone REA599), VioGreen (VG)-conjugated anti-HLA-DR (clone REA805), PE-Vio^®^ 770-conjugated anti-CD86 (clone REA568), PE-conjugated anti-CD45 (clone REA747), PE-conjugated anti-CD40 (clone REA733), APC-Vio 770-conjugated anti-CCR7 (clone REA546), VB and PE-Vio 770-conjugated anti-CD11c (REA618), FITC-conjugated anti-CD209 (clone REA617), VB and FITC-conjugated anti-CD3 (clone REA613), APC-Vio 770-conjugated anti-CD4 (clone REA623), APC-conjugated anti-CD8 (clone REA734), PE-conjugated anti-CD16 (clone REA423). PE-Vio 770-conjugated anti-CD19 (clone REA675), and FITC-conjugated anti-CD54 (clone REA196) were purchased from Miltenyi Biotec.

### 2.2. CD34+ Human HSC Lentiviral Transduction and Expansion

CD34+ cells were thawed in a dry bath at 37 °C, then seeded at 5 × 10^5^ cells/mL in non-treated tissue culture flasks (Corning, Corning, NY, USA) and CellGenix^®^ GMP SCGM supplemented with 10% StemSpan™ CD34+ Expansion Supplement and were cultured for 8 days. Cell culture medium was replenished every 2–3 days while maintaining seeding density. Purity of the CD34+ population was assessed by Flow Cytometry.

Lentiviral vectors (LVVs) with CD93, CD40L, and CXCL13, separated by P2A and T2A linkers, were purchased from Vector Builder (Chicago, IL, USA) and used to transduce the CD34+ cells. The vector encoding the three transgenes has an EFS promoter and ampicillin sequence for antibiotic resistance. The control LVV plasmid vector (used to transduce the lentiviral control DCs) contains EF1A promoter but does not encode three genes of interest. In addition, vectors carrying only one of the three transgenes were used to produce single transgene-expressing DCs. These single transgene vectors had EFS promoters. The lentiviral vector transductions were performed as described [36] and cells were cultured for 5 days post-transduction.

### 2.3. Differentiation of CD34+ Human HSCs to DCs

Following expansion, genetically modified CD34+ cells were harvested and seeded into tissue culture (TC)-treated flasks (Corning) at a density of 5 × 10^5^ cells/mL to initiate monocyte differentiation by culturing with IMDM containing 5% human serum, 35 U/mL rh-SCF, 4000 IU/mL rh-MC-SF, 70 IU/mL rh-IL-3 and 42 U/mL rh-Flt-3L for 11 days. The medium was replenished every 2–3 days, and cells were split once the cell confluency of each flask reached 70%. Flow cytometry was performed to evaluate the surface marker expression and assess the directed differentiation of CD34+ cells into monocytes.

Monocytes were seeded at a density of 5 × 10^5^ cells/mL and cultured in AIM V medium supplemented with 2% human serum, 500 IU/mL rh-IL4 and 500 IU/mL rh-GMC-SF for 3 days to differentiate to imDCs. These imDCs were cultured in AIM V medium supplemented with 2% human serum, 500 IU/mL rh-IL-4, 500 IU/mL rh-GMC-SF, 1400 IU/mL rh-IL-6, 1700 IU/mL rh-IL-1b, 600 U/mL rh-IL-12, 3800 U/mL rh-IL-15, 750 IU/mL rh-TNF-α, and 2 µg/mL PG-E2 for 24 h to differentiate to mDCs. Cells were collected post ice-cold 10 mM EDTA-PBS treatment for 10 min. Cells were cryopreserved (10^7^ cells/mL) in Cryostor 10 (CS10) freezing medium (Biolife Solutions, Bothell, WA, USA) and stored in liquid nitrogen. Phenotyping of the cells was performed by flow cytometry analysis to ensure purity. The mDCs expressing lentivirally delivered CD93, CD40L, and CXCL13 are defined as engineered DCs.

### 2.4. Lysate Generation

MP2 cells and MP2 tumor tissue samples were used to generate lysate to pulse the DCs. From the tissue culture flasks, MP2 cells were harvested, washed with PBS, and resuspended in 10 mL serum-free DMEM at 10^6^ cells/mL. Cells were subjected to 10 cycles of freezing (−80 °C) and thawing (water bath; 56 °C) [37]. Flash frozen MP2-derived tumor tissue samples were thawed in the water bath at 37 °C and were then finely minced with a scalpel before consolidating contents to a 15 mL conical tube. Samples were washed with 1.5 mL of serum-free AIM V and were centrifuged at 300× *g* for 5 min. Supernatant was removed and the tumor tissue samples were suspended in 30 µL of serum-free AIM V per 10 mg of tissue. Samples were then sonicated (20 kHz frequency) for 10 s followed by a 10 s rest repeatedly for a total of 4 min. The samples were then subjected to 5 freeze/thaw cycles [37]. After the freeze/thaw cycles, samples were analyzed for viable cells to ensure complete lysis. The samples were spun down at 1000× *g* for 5 min at 4 °C to collect lysate supernatant and stored at −80 °C in aliquots for future use. Total protein in the lysate was then determined by Detergent Compatible protein assay (Cat# 5000112, Bio-Rad, R&D Systems, Minneapolis, MN, USA).

### 2.5. Antigen Pulsing of DCs

mDCs were pulsed with either MP2 cells or tumor lysate by following previously published procedures [37,38]. For each 10^6^ mDCs, 18 µg of lysate protein was added. After 18 h of incubation, the mDCs were harvested, washed, and cryopreserved as previously described above. After antigen pulsing, the resulting DCs are defined as antigen-pulsed engineered DCs.

### 2.6. Flow Cytometry

Immunophenotyping of cells was performed using flow cytometry. Surface staining was performed by utilizing REA mAbs, as per the manufacturer’s recommendations. Cell viability was evaluated using propidium iodide or 7AAD (Miltenyi). Flow cytometry was performed using Miltenyi MACSQuant Analyzer 10 or Beckman Coulter Epics XL Cytometer (Brea, CA, USA). FlowJo software (v10.10.0) was used to analyze data.

### 2.7. Enzyme-Linked Immunosorbent Assay (ELISA)

ELISA experiments were performed as previously described [39]. Quantification of secreted CD93, CD40L, and CXCL13 from the lentiviral transduced cells was performed using Human C1qR1/CD93 Quantikine ELISA Kit (R&D Systems, Cat# DCD930), Human CD40 Ligand/TNFSF5 Quantikine ELISA Kit (R&D Systems, Cat# DCDL40), and CXCL13/BLC/BCA-1 ELISA Kit (ThermoFisher, Cat# EHCXCL13, Waltham, MA, USA) by following manufacturers’ recommendations. Varioskan Lux plate reader (ThermoFisher) was used to read ELISA plates.

### 2.8. Vector Copy Number and RT-qPCR

To assess lentiviral integration to target cells, vector copy number (VCN) was quantified using qPCR. Genomic DNA was isolated from pre- and post-transduction cells with either the LVV or control LVV, using the AllPrep DNA/RNA Mini Kit (Qaigen, Cat# 80204, Hulsterweg, The Netherlands) according to the manufacturer’s protocol. DNA concentration and purity were assessed using a NanoDrop One Spectrophotometer (Thermo Scientific) and DNA agarose gel electrophoresis. qPCR reaction was performed on a CFX96 Real-Time System (Bio-Rad) using TaqMan Universal Master Mix II (Thermo Scientific, Cat# 4440040) as per manufacturers’ recommendations. The cycling conditions were as follows: polymerase activation at 95 °C for 10 min, followed by 40 cycles of denaturation at 95 °C for 15 s, annealing and extension at 60 °C for 1 min. The primer and probe set were designed and purchased from IDT (Newark, NJ, USA). The following sequences (5′ to 3′) include the forward primer, backward primer, and probe, respectively: TCCTGGCCTGTTAGAAACATCAG, GCACACAATAGAGGGTTGCTAC,/56-FAM/CTGGGACAG/ZEN/CTACAACCATCCCTT/3IABkFQ/. This primer probe set was designed to target the lentiviral backbone sequence which includes both the LVV and control LVV, irrespective of transgene sequence presence. Amplification of the reference gene RNase P was used to determine the number of diploid genomes, while amplification of a known lentiviral plasmid was used to quantify vector copies. Serial dilutions of both the reference gene and the lentiviral plasmid were performed to generate standard curves, allowing the calculation of copy numbers from the Cq values of the experimental samples.

To determine transgene expression at the transcript level, RT-qPCR was performed. Total RNA was isolated from pulsed DCs using AllPrep DNA/RNA Mini Kit (Qaigen, Cat# 80204) according to the manufacturer’s protocol. RNA concentration and purity were assessed using a NanoDrop One Spectrophotometer (Thermo Scientific) and agarose gel electrophoresis. RT-qPCR was performed on a CFX96 Real-Time System (Bio-Rad) using the PrimeTime One-Step RT-qPCR Master Mix (IDT, Cat# 10007066) as per the manufacturers’ recommendations. The cycling conditions were as follows: reverse transcription at 50 °C for 15 min, polymerase activation at 95 °C for 3 min, followed by 40 cycles of denaturation at 95 °C for 15 s and annealing and extension at 60 °C for 1 min. Cq was determined using the software provided by the CFX96 Real-Time System. The relative expression levels of the transgenes were calculated as previously described [40] by normalizing to the level of the reference gene, GAPDH. A detection threshold of 35 cycles was used. The following primers and probes (IDT) were used to detect genes of interest:

CD93 (sequence 5′ to 3′)

Forward: TTCCAGAGCGAGCTGAGAG

Reverse: ACAGAGAGAAGTTCGTGGCT

Probe:/56-FAM/TACAGTCCG/ZEN/ACACCTGGGACAGA/3IABkFQ/

CD40L (sequence 5′ to 3′)

Forward: GACTGATCCAAGCCAAGTGAG

Reverse: CCGCATGTTAGAAGACTTCCC

Probe:/56-FAM/TGGCTTCAC/ZEN/GTCCTTTGGCTTACT/3IABkFQ/

CXCL13 (sequence 5′ to 3′)

Forward: TCAAGCTGAATGGATACAAAGAATG

Reverse: CTTTGTACAAGAAAGCTGGGT

Probe:/56-FAM/AGTTCTTCA/ZEN/ACTCTACCAGTTCCAGTGT/3IABkFQ/

### 2.9. Orthotopic Tumor Implantations and Treatment with DCs

Male and/or female Hu-BLT or Hu-PBMC-bearing NSG mice, were obtained from the University of California, Los Angeles (UCLA) animal facility and were housed in pathogen-free, relative humidity of 45 to 65% at a temperature between 18 °C and 22 °C. Mice received sterilized food and water ad libitum and 12 h light/dark cycles. Handling of the mice was performed under the written approval of the UCLA Animal Research Committee (ARC) in accordance with all federal, state, and local institutional guidelines. The UCLA Division of Laboratory Animal Medicine (DLAM) oversaw the monitoring of mice. Orthotopic pancreatic tumors were implanted as previously described [41]. MP2 cells were prepared for injection by suspending 10^6^ cells in 7 µL of Matrigel (Corning) and kept on ice. Each mouse was placed in an induction chamber with the air or O_2_ flow meter set to 1 L/min and the isoflurane vaporizer set to 3% to sedate and then transferred to the surgical field on its back with a nose cone. Ointment was applied to the eyes to prevent corneal dehydration. The iodine solution was applied in a circular motion on the abdomen. An incision was made in the upper left quadrant of the abdomen and expanded to 2 cm. The peritoneal cavity was exposed, and the intestines were moved to locate the spleen. The body of the pancreas was gently exposed, and 100 µL cell suspension was slowly injected with a 27-gauge insulin syringe. A sterile gauze pad was held over the injection site for 30 s to prevent leakage. 4–0 silk sutures were used to sew the peritoneum and skin. 200 µL of Carprofen (MilliporeSigma, Burlington, MA, USA) was injected at 1 mg/kg post-surgery at 24 h and 48 h to manage pain and inflammation. Based on previous studies [41], in this experiment we used 32 mice in total (23 Hu-BLT, 9 hu-NSG) distributed randomly as follows: *n* = 7 for each control group and *n* = 9 for engineered DCs for Hu-BLT, and *n* = 3 for each control group and *n* = 3 for engineered DCs for HU-NSG. After one and two weeks, mice were dosed with 10^6^ antigen-pulsed DCs via intradermal (i.d.) injection next to the surgical wound. Substantial evidence suggests that this injection route allows the engineered DCs to efficiently migrate to regional lymph nodes, where they can effectively prime antigen-specific T cells [42]. A strategy of two therapeutic doses of engineered DCs, where the second dose was given a week after the first dose, was adopted for this study. This approach was intended to boost the therapeutic efficacy of the treatment regimen, helping to inform optimal dosing for future translational studies. The mice were monitored for tumor growth and overall health for 5 weeks.

### 2.10. PBMC and CD8+ T Cell Isolation

PBMCs were isolated from peripheral blood using the Ficoll-Hypaque method as previously described. The PBMCs were then used to isolate CD8+ T cells using a STEMCELL Technologies EasySep™ CD8+ T-cell Isolation Kit, following the manufacturer’s protocol. The purity and viability of the cells were assessed using flow cytometry and trypan blue exclusion, respectively.

### 2.11. DC-CD8+ T Cell Co-Culture and In Vitro T Cell Activation of CD8+ T Cells with DCs

CD8+ T cells were co-cultured with DCs that had been previously thawed and cultured for 24 h at a 1:1 ratio (typically 10^6^ cells of each type) in a 12-well plate in a final volume of 1 mL of AIM V medium (Gibco) supplemented as described in Section 2.3. After the co-culture, the cells were harvested, and the supernatant was collected by gentle centrifugation at 575× *g* for 5 min to separate the cells from the culture medium. The supernatant was stored at −20 °C for subsequent quantification of IFN-γ by ELISA (BioLegend Systems), following the manufacturer’s protocol. The IFN-γ levels were measured to assess the activation status of CD8+ T cells in response to DC priming. After supernatant collection, the CD8+ T cells were counted using a hemocytometer ensuring that their viability was assessed through trypan blue exclusion. The harvested CD8+ T cells were then used as effector cells for a cytotoxicity assay.

### 2.12. ^51^Cr Release Cytotoxicity Assay

The ^51^Cr (chromium-51) release assay was performed as previously described [43]. Briefly, different ratios of PBMCs or NK cells and ^51^Cr-labeled OSCSCs were incubated for 4 h. Afterward, the supernatants were harvested from each sample, and the released radioactivity was counted using the gamma counter. Lytic units (LU) 30/10^6^ were calculated by using the inverse of the number of PBMCs or NK cells needed to lyse 30% of tumors ×100.

### 2.13. Cytotoxicity Analysis by e-Sight System

The RTCA eSight system (Agilent, Santa Clara, CA, USA) was employed to investigate cell behavior and function using real-time impedance-based and image-based measurements. This system utilizes specialized E-Plates embedded with gold biosensors at the bottom of each well, enabling non-invasive quantification of cellular events such as cell adhesion, proliferation, and cytotoxicity. On Day 1, 50 µL of supplemented DMEM was added to each well to measure background impedance. Following this, another 50 µL of DMEM containing 10,000 target tumor cells (MP2-GFP or SNU-GFP cells) was seeded into the E-Plates. The plates were incubated at 37 °C with 5% CO_2_ and continuously monitored by the RTCA system and their adhesion and proliferation were tracked for 24 h via impedance-based and image-based measurements. Impedance readings were recorded every 5 min, and images were captured every 60 min to track tumor cell adhesion and proliferation. After 24 h, 100 µL of supplemented RPMI containing effector cells (Activated CD8+ T cells post-DCs co-culture) were added to each well in serial dilutions to achieve different effector:target cell (E/T) ratios. The highest ratio used was 2.5:1, where 12,250 effector cells were added per well. Subsequent dilutions included ratios of 1.25:1 (6125 effector cells). Impedance measurements were recorded every 15 min, and images were captured every 60 min for a period of 40 h to monitor effector–target interactions, cytotoxicity, and cell behavior in real time, as per the manufacturer’s recommendations.

### 2.14. Immunohistochemical Analysis and Digital Pathology

IHC was performed as previously described [44]. Briefly, tissue sections of 4 µm were dewaxed in xylol (J.T. Baker-Avantor, Radnor, PA, USA) and rehydrated through a series of reagents (xylene, 100% ethanol, 90% ethanol, 70% ethanol and distilled water). Antigen retrieval was performed. Endogenous peroxidase activity was blocked in a 10% hydrogen peroxide solution (J.T. Baker), and slides were incubated with specific primary antibodies (anti-CD4 or anti-CD8). Sections were incubated with the ImmPRESS HRP Horse Anti-Rabbit IgG Polymer Detection Kit Peroxidase from Vector Laboratories (Burlingame, CA, USA) and DAB detection system (Vector Laboratories), counterstained with hematoxylin, dehydrated, and mounted.

The IHC-stained sections were digitized using an Aperio ScanScope CS (Leica Biosystems, Germany), which obtains 20X digital images with high resolution (0.45 μm/pixel). Images were viewed using an ImageScope (Aperio, San Diego, CA, USA) to analyze and quantify marker expression. An algorithm was developed for each tissue to quantify total and nuclear protein expression. The ImageScope allows the setting of a threshold for color saturation as well as upper and lower limits for intensities of weak, moderate, and strong positive pixels. To ensure settings were appropriate for analysis of different tissues, randomly selected cores on the parameters were customized to differentiate between negative (blue), low (yellow), moderate (orange), and strong (red) staining. The raw data included the number of positive pixels and the intensity of positive pixels, which were normalized to the number of total pixels counted in µm^2^. Data is presented as total density/µm^2^ analyzed in a total area of 10,000 µm^2^.

### 2.15. Statistical Analysis

Student’s t test (two groups) or ANOVA (analysis of variance) and Tukey’s test (three or more groups) were performed for statistical analyses (****—*p* value < 0.0001, ***—*p* value < 0.001, **—*p* value < 0.001–0.01, and *—*p* value < 0.01–0.05). For non-normal data, the Mann–Whitney U test was performed for two-group contrasts. Bonferroni correction was used for multiple comparisons (two genes) with adjusted α = 0.05/2 = 0.025. All statistical analyses were performed using the GraphPad Prism-10.3.0.507 software. All schematic figures were generated using BioRender (2.0.1).

## 3. Results

### 3.1. Derivation of Engineered CD11c+ DCs from CD34+ HSCs

To derive engineered DCs expressing CD93, CD40L, and CXCL13, CD34+ human hematopoietic stem cells (HSCs) were thawed and expanded as illustrated in Figure 1A. Flow cytometry analyses of the starting material showed that >98% of cells were positive for the CD34+ marker. All the cells expressed the CD45 marker at all points of analysis (Figure 2 and Appendix A). These CD34+ HSCs were transduced with lentiviral vectors (Appendix A) encoding human CD93, CD40L, and CXCL13 or a control lentiviral vector with a stuffer sequence. The transduced cells were subjected to a two-step differentiation process; first to derive CD14+ monocytes, followed by further differentiation to generate CD11c+ DCs (Figure 1A). After the lentiviral transduction, CD34+ HSCs were further expanded in SCGM (Sartorius CellGenix, Germany) supplemented with StemSpan™ CD34+ Expansion Supplement (Stem Cell Technologies, Canada). The expressions of CD93 and CD40L were used as markers to assess transduction efficiency. Flow cytometry results were analyzed utilizing the gating strategy detailed in Figure 2A. The flow cytometry analyses post-transduction revealed that 38% and 18% of cells were CD93+ and CD40L+, respectively (Appendix A).

**Figure 1 vaccines-13-00749-f001:**
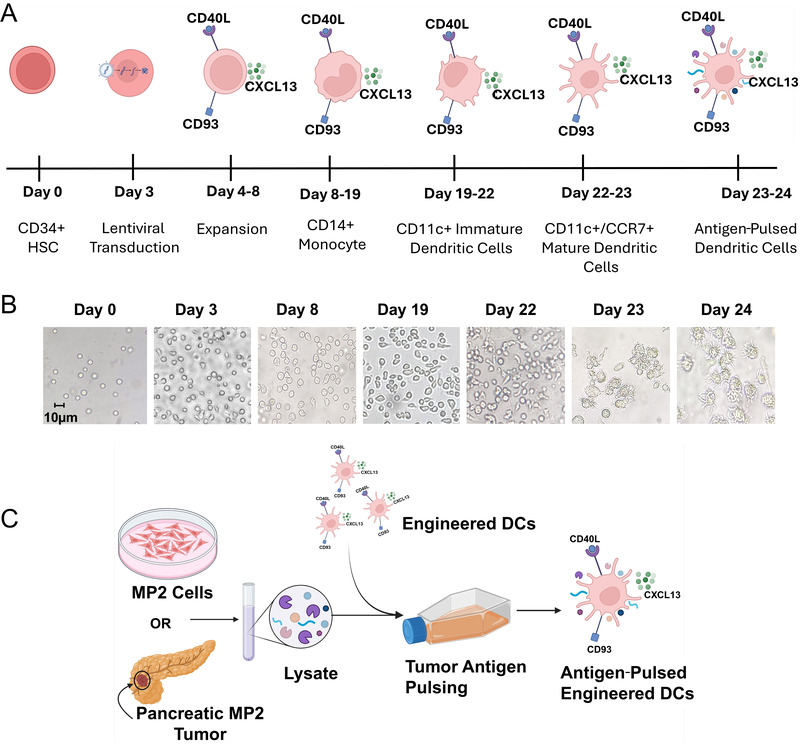
Production of engineered DCs. (**A**) Schematic of CD34+ cell differentiation to engineered DCs. (**B**) Representative brightfield images of cells at each stage of the differentiation process. Scale = 10 µm. (**C**) Schematic of the MP2 cell and tumor antigen sourcing for use in pulsing of engineered DCs.

By the end of the HSC expansion, cells exhibited downregulation of the CD34+ marker from 98.9% to 28.9% (Figure 2B). After the expansion, the lentiviral-transduced cells were subjected to monocyte differentiation in the presence of recombinant human (rh)-SCF (stem cell factor), rh-MC-SF (macrophage colony-stimulating factor), rh-IL-3 (interleukin 3), and rh-Flt-3L (Fms-related tyrosine kinase 3 ligand) from Day 8 to Day 19. As the cells progressed through the differentiation process, the downregulation of the CD34 marker was observed from 98% on Day 3 to less than 2% on Day 19 (Figure 2B). This loss of the CD34 marker corresponded with the initiation of monocyte-specific cell-surface marker expression. This reduction in CD34+ surface marker and upregulation of CD14+ is correlated with the changes in cellular morphology (Figure 1B). By the end of monocyte differentiation and expansion (on Day 19), 22.8% of cells were positive for the monocyte-specific marker CD14 and a low percentage of cells were positive for DC-specific markers such as CD86 (19.2%), CD40 (15.1%), CCR7 (12.0%), CD11c (15.6%), and human leukocyte antigen (HLA)-DR (33.0%), while 99.3% of cells were positive for HLA-ABC (Figure 2C–E, Figure 3 and Appendix A). From this point onwards, only <1% of cells were positive for the CD34+ marker (Appendix A). From Day 19 to Day 22, monocytes (CD14+) were further differentiated to imDCs by culturing them in the presence of rh-IL4 (recombinant human Interleukin-4) and rh-GM-CSF (granulocyte-macrophage colony-stimulating factor). This differentiation step resulted in the upregulation of DC-specific surface markers such as CD86 (23.2%), CD40 (20.9%), CCR7 (13.3%), CD11c (20.9%), and HLA-DR (45.4%) (Figure 2C–E, Figure 3 and Appendix A). These imDCs were further differentiated by culturing them in the presence of rh-IL-4, rh-GMC-SF, rh-IL-6, rh-IL-1b, rh-IL-12, rh-IL-15, rh-TNF-α, and PG-E2 (prostaglandin-E2) for 24 h to derive mDCs as confirmed by the cell surface marker expression profile: CD86 (50.0%), CD40 (57.8%), CCR7 (44.3%), CD11c (88.3%), HLA-DR (49.8%), and an adherent cell phenotype.

**Figure 2 vaccines-13-00749-f002:**
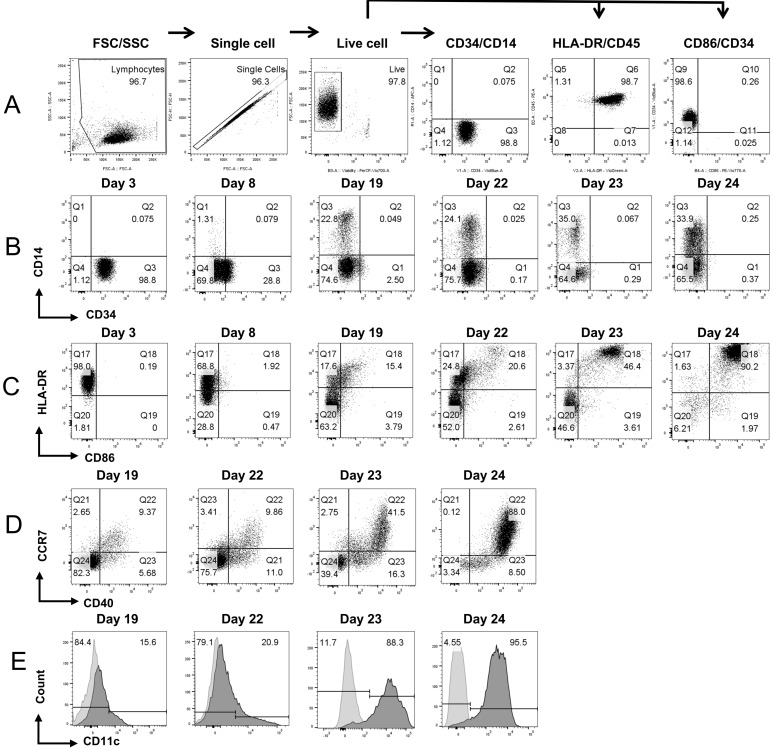
Surface marker analysis by flow cytometry. (**A**) Flow cytometry gating strategy. Representative histograms for the expression of the following surface markers: (**B**) CD14 and CD34; (**C**) HLA-DR and CD86; (**D**) CCR7 and CD40; and (**E**) CD11c.

Antigen pulsing was performed by culturing the mDCs in the presence of cytokines rh-IL-4, rh-GMC-SF, rh-IL-6, rh-IL1b, rh-IL12, rh-IL-15, and rh-TNF-α and incubating 10^6^ cells with 18 µg of tumor lysate or cultured cell lysate for 18 h (Figure 1C). This antigen pulsing process correlated with the upregulation of key surface markers that are considered crucial for the antigen presentation function of the DCs, such as CD86 (92.2%), CD40 (96.5%), CCR7 (88.1%), CD11c (95.5%), and HLA-DR (91.8%) (Figure 2C–E, Figure 3, Appendix A). The final drug product and/or the engineered DCs were also analyzed and confirmed for purity by measuring low/absent expression of CD34, CD3, CD4, CD8, CD16, CD19, and CD56. Data showed the absence of these markers in the engineered DCs, with the exception of some expression in CD4 (Appendix A). These flow cytometry results are illustrated in Figure 3A,B and showcase the upregulation of APC functional markers upon CD14+ monocyte differentiation and in CD11c+ engineered DCs. In our process, for each CD34+ cell, we were able to produce an average of 60 ± 28 (*n* = 5) DCs. The resulting cells transduced with the LVV stuffer are classified as LVV control DCs, the cells transduced with the LVV encoding CD93, CD40L, and CXCL13 are classified as engineered DCs, and the unmodified DCs produced in the same method are classified as untransduced (UTD) control DCs. In addition, DCs expressing either CD93 or CD40L or CXCL13 were also produced in the same manufacturing process to demonstrate the capabilities of these individual genes versus those in combination.

**Figure 3 vaccines-13-00749-f003:**
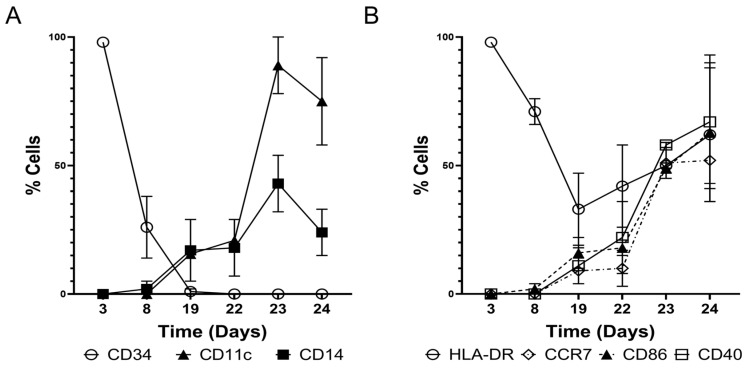
Cell surface marker analysis of CD34+ HSC to DC differentiation. (**A**) Representative histograms for the expression of CD34+, CD14, and CD11c (Day 3 vs. Day 24: CD34 *p* < 0.0001, CD14: *p* < 0.05, CD11c *p* < 0.05) (**B**) HLA-DR, CD40, CD86, and CCR7 (Day 3 vs. Day 24: CD40: *p* < 0.05, CCR7 *p* < 0.01). Data is presented as mean ± standard deviation (*n* = 3).

### 3.2. Engineered DCs Express CD93, CD40L, and CXCL13

To analyze lentiviral integration, total DNA was isolated from CD34+ HSCs (pre-transduction), UTD control DCs, lentiviral control DCs, and engineered DCs. The primer/probe set was designed to specifically target the vector sequence, and quantitative real-time PCR (qPCR) was employed. As expected, the LVV sequences were detectable in both the lentiviral control DCs and the engineered DCs, indicating lentiviral integration into the genome, while absent in the UTD control DCs and CD34+ HSCs (Figure 4A). While the LVV control DCs exhibited lower vector copy numbers (VCNs) compared to engineered DCs, this difference was not statistically significant. The mechanism underlying such minor non-significant variation is not clear at present.

Next, we evaluated transgene expression using RT-qPCR with primers and probes specifically designed to target lentiviral transgene sequences and not endogenous genomic DNA sequences. The amplification curve generated during the PCR reaction illustrated the detection of the transgenes in the engineered DCs with quantification cycle threshold (Cq) values ranging from 23–25.5 cycles (Appendix A). In contrast, the samples from the lentiviral control DCs (Appendix A) had no amplification for transgenes, indicating the absence of transgenes in these groups. In addition, the relative gene expression of these transgenes was determined by normalizing GAPDH (Figure 4B), and the data showed mRNA transgene expression in the engineered DCs.

To demonstrate protein expression of CD93, CD40L, and CXCL13 transgenes, enzyme-linked immunosorbent assay (ELISA) and flow cytometry assays were employed. Culture supernatants from pre- and post-transduction were analyzed for the presence of lentiviral-encoded transgene proteins by ELISA. In cell culture supernatants of CD34+ HSCs pre-transduction, CD40L and CXCL13 expression was undetectable by ELISA. Post-transduction, cell culture supernatants of the engineered DCs showed an increase of transgene expression and were positive for CD93, CD40L, and CXCL13 expression by ELISA, where cells were on average secreting 2873.91 pg/mL of CD93, 33.99 pg/mL of CD40L, and 44.24 pg/mL of CXCL13 (Figure 4C, left, middle, and right, respectively), whereas cell supernatants derived from the LVV control DCs were undetectable by ELISA for CD40L and CXCL13 (Figure 4C, middle and right) and showed lower expression of CD93 at 298.88 pg/mL, in comparison to the engineered DCs (*n* = 3) (Figure 4C, left). In addition to RNA expression and identifying transgene-encoded proteins in the cell culture supernatant, surface expression of CD93 and CD40L was tested by flow cytometry analyses. Engineered cells exhibited a 2.55-fold increase for CD93 and a 1.83-fold increase for the expression of CD40L compared to LVV control samples (Figure 4D).

### 3.3. Engineered DCs Significantly Enhanced Effector Cell Activation In Vitro

To demonstrate the efficacy of engineered DCs in enhancing the activation of effector cells, we first co-cultured PBMCs with either pulsed or unpulsed engineered DCs and LV control DCs. Interferon-gamma (IFN-γ) levels were measured in the culture supernatants after 5 days. Notably, co-culturing of T cells with the pulsed engineered DCs resulted in significantly higher IFN-γ secretion (313.26 pg/mL) than the unpulsed or control groups (133.3 pg/mL) (Figure 5A), indicating that antigen presentation plays a crucial role in the activation of PBMCs.

To assess the cytotoxic activity of CD8+ T cells following the co-culture with MP2 lysate-pulsed DCs, we conducted a cytotoxicity assay using the xCELLigence Real-Time Cell Analysis technology (RTCA) eSight system. The target tumor cells MIA PaCa-2, expressing green fluorescent protein (MP2-GFP), were seeded and allowed to adhere for 24 h before the addition of effector CD8+ T cells at a 1.25:1 ratio (effectors: target (E/T ratio)). The impedance measurements were recorded every 15 min and images were taken every 60 min for 40 h. The results demonstrated that co-culture with MP2 lysate-pulsed engineered DCs led to the highest percentage (43%) of lysis of target cells compared to CD8+ T cells co-cultured with UTD control DCs (0%), LVV control DCs (5%), CD93-DCs (14%), CD40L-DCs (0%), and CXCL13-DCs (0%) (Figure 5B). These findings highlight the enhanced ability of engineered DCs to activate CD8+ T cells, resulting in improved tumor cell killing. Additionally, representative images captured at the 40 h time point illustrate the morphological differences and interactions between the effector and tumor cells in each DC treatment group (Figure 5C). To further support this, we evaluated the specificity of the cytotoxic function of CD8+ T cells that had been previously co-cultured with DCs pulsed with pancreatic tumor lysate. These CD8+ T cells were able to specifically kill pancreatic tumor cells but not liver carcinoma cells (SNU 423-GFP) (less than 2% of cytolysis) (Figure 5D). This result indicates that the activation and cytotoxic activity of CD8+ T cells was tumor antigen-specific rather than the consequence of a generalized alloimmune reaction. In addition, the results showed that there was no cytotoxic effect observed against SNU 423-GFP cells by CD8+ T cells that were co-cultured with DCs expressing LVV control or UTD control, indicating the specificity of the engineered DC treatment (Figure 5D).

### 3.4. Treatment with Engineered DCs Leads to Pancreatic Tumor Regression in the hu-BLT Mouse Model

To evaluate the effectiveness of engineered DCs’ capacity to induce an anti-tumor effects in vivo, studies involving hu-BLT mouse model were conducted. Hu-BLT mice were inoculated with 10^6^ human MP2 cells through aseptic surgery directly into the pancreas (Figure 6A) and were randomly divided into three experimental groups: the first group consisted of untreated tumor-bearing mice and served as a control group, the second group received 10^6^ LVV control DCs, and the third group received 10^6^ engineered DCs (pulsed with MP2 lysate). DCs were administered intradermally on Day 7 and on Day 14 post-tumor implantation. The 35-day experimental endpoint was selected to provide assessment of therapeutic efficacy while considering practical constraints such as animal welfare, model stability, and resource availability. This timeframe allowed us to capture early tumor responses and immune activation in humanized mouse models. After 35 days, mice were sacrificed, tumors were collected, and the tumor weight and volume measurements were taken. Data from three independent studies demonstrated a significant reduction in tumor size in mice treated with engineered DCs (70.46% decrease) compared to untreated controls and mice receiving LVV control DCs (Figure 6B,C). Similarly, treatment with MP2 lysate-pulsed engineered DCs resulted in an 81.7% reduction in tumor volume compared to untreated controls (Figure 6C, right panel). A significant decrease in tumor volume was also observed in the group of mice that received the LVV control DCs (41.69%), albeit less successfully compared to the group with the tumor untreated.

Additionally, the therapeutic efficacy of the engineered DC treatment was assessed using the hu-PBMC mouse model with orthotopic pancreatic tumors. Like the Hu-BLT mouse model, the treatment with MP2-lysate pulsed engineered DC treatment resulted in a significant reduction in tumor growth compared to untreated control and LVV control DCs groups (Figure 6D). When we evaluated the tumor weight and volume, the data revealed a 40% decrease in tumor weight and a greater than 70% decrease in tumor volume (Figure 6E). These results confirmed the therapeutic efficacy of engineered DC treatment in pancreatic tumor-bearing mouse models. Statistical power analysis details for this experiment, denoted in Figure 6, are described in Appendix A.

### 3.5. Engineered DCs Promote Tumor-Targeting Immune Responses In Vivo

Immune cell activation was evaluated in pancreatic tumor-bearing mice upon treatment with either the engineered DCs, LVV control DCs, or untreated controls using peripheral blood, spleen, and bone marrow (BM) samples. First, we analyzed the cell population of CD4+, CD8+ T cells, and NK cells in peripheral blood samples using flow cytometry. We observed a statistically significant increase in the number of CD4+ T cells (39.0%), CD8+ T cells (10.7%), and NK cells (2.9%) in PBMCs in the group of mice that received the antigen-pulsed engineered DCs (39%, 10.7%, and 2.9%) compared to the control groups (Figure 7A). Similar results were observed in the BM samples from the antigen-pulsed engineered DCs, showing an increase in CD4+ T cells (22.4%), CD8+ T cells (1.28%), and NK cells (3.79%) compared to control groups (Figure 7B).

Next, we evaluated the IFN-γ expression in purified splenocytes and BM cells upon treatment with the engineered DCs or control DCs. In these experiments, splenocytes and BM cells were further activated in vitro with IL-2 or IL-2+aCD3/28, or IL-2+ the probiotic sAJ4 for 24 h and assessed for IFN-γ release. Engineered DC treatment resulted in splenocytes and BM cells expressing higher levels of INF-γ compared to the samples from the mice treated with UTD control DCs and LVV control DCs (Figure 7C,D). This IFN-γ release was enhanced following treatment with engineered DCs in both splenocytes (113.3 pg/mL, 608.8 pg/mL, and 1622.1 pg/mL) and BM (923.0 pg/mL, 1221.0 pg/mL, and 1187.0 pg/mL) cells upon treatment with IL-2, IL2+aCD3/28, and sAJ4. To test the cell-killing ability of these IFN-γ-releasing splenocytes, these cells were then incubated with chromium-51-labeled OSCSCs and chromium release assay was performed. The results revealed that splenocytes derived from engineered DC treatment resulted in higher cytotoxicity when compared to the splenocytes derived from mice treated with control DCs (Figure 7E).

### 3.6. Enhanced Tumor Infiltration of CD4+ and CD8+ T Cells Following Antigen-Pulsed Engineered DC Treatment

To understand the mechanism behind pancreatic tumor regression upon treatment with engineered DCs, we employed immunohistochemistry (IHC) analysis to assess the tumor infiltration of CD4+ and CD8+ T cells. Pancreatic tumor samples from mice treated with either engineered DCs, LVV control DCs, or untreated controls were collected, and paraffin-embedded sections were stained using anti-CD4 and CD8 antibodies. Digital pathology techniques were employed to quantify the presence of CD4+ and CD8+ cells within the TME. IHC analysis demonstrated a significant increase (705.1 cells/µm^2^) in the tumor infiltration of CD8+ T cells upon treatment with engineered DCs compared controls (tumor alone), and normal tissue (Figure 8A,B). These findings suggest that engineered DCs play a crucial role in enhancing the infiltration of cytotoxic CD8+ T cells into pancreatic tumors, potentially contributing to improved anti-tumor immunity.

## 4. Discussion

PC is complex and presents multiple challenges that hinder effective treatment. It is often asymptomatic and difficult to diagnose at an early stage. For most patients, diagnosis occurs at a late stage, with numerous metastatic lesions and poor overall survival [45]. DCs are the key APCs that can activate both innate and adaptive immunity and present a potentially effective treatment of solid tumors [13,46]. In this paper, we present a strategy for engineering CD34+ cells to derive mDCs, overexpressing CD93, CD40L, and CXCL13, followed by antigen pulsing for enhanced targeting of pancreatic tumors.

One of the limitations for generating gene-modified DC-based therapies resides in the difficulty of effective gene delivery to DCs and their relative scarcity in peripheral blood [47]. For this reason, CD34+ HSCs proved to be an ideal starting material due to their capacity to expand to large numbers, differentiate to intermediate immune-stimulating monocytes, and ease of engineering through lentiviral transduction [48]. The production of engineered DCs presented several challenges. To address these, a robust strategy was implemented, focusing on critical quality control checkpoints at key stages of the manufacturing process, as illustrated in Figure 1A. These checkpoints coincided with major differentiation steps and utilized assays, such as flow cytometry and gene expression analysis, to confirm successful guided differentiation. If any checkpoint revealed the absence of essential markers the manufacturing process would be halted and new production would be initiated using new starting materials. This rigorous approach ensures high reproducibility across multiple production batches, enabling scalable and consistent output of engineered DCs.

Our data demonstrate that CD34+ expression is retained during initial HSC expansion followed by their differentiation into CD14+ monocytes. As monocytes differentiate into imDCs, we observed an upregulation of MHC-II (HLA-DR), CD40, and CD86, confirming the imDC phenotype. Continued differentiation led to increased expression of CD86, CD40, CCR7, and CD11c, indicating the generation of mDCs. Morphologically, the cells display uniform, dendrite-like structures. Upon antigen pulsing, these engineered DCs show enhanced surface expression of key immunostimulatory markers. The upregulation of these key APC functional proteins is critical for the uptake of tumor-associated antigens and eliciting immunomodulatory responses, resulting in elevated T cell functionality [49]. In addition, CCR7 not only confirms the mDCs phenotype but is also a critical chemokine receptor necessary for the migration of tumor-infiltrating DCs into the tumor-draining lymph node (TDNL) and may also be involved in the recruitment of DCs into the TME [50,51]. We acknowledge that 34.2% of the final engineered DC still expresses CD14, and this could be due to the transient expression of the surface marker that is common in CD34-derived DCs and may even synergize with naturally occurring DC subsets [52,53].

The three key genes, CD93, CD40L, and CXCL13, were selected for their important physiological function to increase binding, recruitment, and co-stimulation of T cells [21,33,54]. A tri-cistronic lentiviral vector with 2A linking peptides was constructed to co-express these genes simultaneously [55]. The three genes were cloned as CD93-P2A-CD40L-T2A-CXCL13 with an EF1 Alpha Short (EFS) promoter (Appendix A). In general, the levels of CD40L were lower than the other two genes. This could be potentially due to the nature of protein expression in a tri-cistronic construct showing that the lowest protein expression (up to 70% decrease) levels were at the second position [56]. While we observed variations in the expression levels of the three genes, the specific reasons for these differences remain unclear. Further studies will be conducted to investigate the underlying mechanisms contributing to this variability. In this study, we are reporting on a clinically proven technology that has been used for gene editing studies that were approved by the FDA [57]. Advanced genetic engineering technologies such as RNA electroporation, lipid nanoparticle (LNP) delivery, CRISPR, adeno-associated viral particle delivery, or other methods will be explored to further refine the derivation of engineered DCs in the future.

To assess the role of each transgene, single gene-modified DCs (CD93, CD40L, or CXCL13) and fully engineered DCs (expressing all three) were tested in vitro with CD3^+^ and CD8^+^ T cells. Antigen-pulsed engineered DCs induced significantly higher IFN-γ secretion and tumor cell killing than single-gene DCs, indicating the requirement of CD93, CD40L, and CXCL13 combination for optimal T cell activation and antitumor effects (Figure 5). These transgene-expressing DCs may also activate broader immune pathways, including B and NK cell responses. Additionally, the specificity of engineered DC-derived T cell activation was confirmed using unrelated cancer cells (SNU-GFP). While UTD DCs can induce some immune responses, engineered DCs demonstrate superior and antigen-specific immune modulation. Furthermore, UTD control DCs were used during the in vitro experiments illustrated in Figure 5B,C; however, based upon these results showing a lack of response initiated by the UTD DCs, remaining studies were conducted without this control group. We acknowledge that the present manuscript does not directly dissect the molecular mechanisms by which the combined expression of CD93, CD40L, and CXCL13 on engineered DCs enhances CD8+ T cell cytotoxicity. Based on the published literature, we propose the following plausible mechanisms: 1. CD40L expression on DCs may promote their own maturation and upregulation of co-stimulatory molecules (e.g., CD80 and CD86), thereby enhancing their ability to prime CD8+ T cells effectively [58]. 2. CD93 (Fas) expression in DCs has been shown to have non-apoptotic roles in modulating the inflammatory environment and promoting DC activation [59], which could contribute to robust T cell priming. 3. CXCL13 production may enhance recruitment and organization of T cells and B cells within lymphoid structures, potentially improving the efficiency of T cell–DC interactions [60], thus favoring CD8+ T cell activation and expansion. Taken together, we believe that these transgenes play key roles in optimal T cell engagement and the immune activation that is required for antitumor activity. The in vitro experiments were the gateway to in vivo studies using humanized mouse models bearing pancreatic tumors. Various mouse models such as GEM, xenograft (NSG), and humanized (Hu-BLT, Hu-PBMC) are used to study cancer progression and treatment [61,62]. In this study, we employed humanized mouse models with human PC to better evaluate the therapeutic efficacy of our allogeneic-engineered DCs. These models, incorporating human immune components, are valuable for preclinical immunotherapy research, though no single model fully replicates human immunity, underscoring the need for complementary approaches [34,62,63].

In this regard, we tested the therapeutic efficacy of the platform in two different humanized mouse models (Hu-BLT and Hu-NSG) to closely represent human disease [34,35]. With the results using both mouse models bearing pancreatic tumors, the administration of engineered DCs leads to therapeutic efficacy that includes reductions in both tumor volume and tumor weight (Figure 6B–E). We observed that only mice treated with engineered DCs pulsed with tumor lysate showed a significant inhibition of tumor growth compared to mice treated with control DCs pulsed with the same tumor lysate. Although LVV control DCs were also able to decrease tumor weight and volume, the levels were lower when compared to the engineered DC treatment, underscoring the added potency of the engineered cells. These results further support the additional functional properties conferred by the engineering of the DCs to overexpress CD93, CD40L, and CXCL13. Results from our study align with previously published work reporting the capacity of the three transgenes in optimizing T cell engagement and enhancing immune system activity and migratory capacity, thus enhancing tumor killing [36,47,64,65]. Engineered DCs were administered intradermally on days 7 and 14 post-tumor implantation, and animals were sacrificed on day 35. Although graft-versus-host disease (GvHD) is a common and potentially life-threatening complication following allogeneic cell transplantation [66], no clinical signs of GvHD were observed within the treatment window. This observation aligns with previous reports indicating that GvHD in the Hu-BLT model typically develops at later stages [67]. However, the biology of chronic GvHD, particularly its autoimmune-like manifestations, remains poorly understood, largely due to the lack of representative long-term preclinical models. Therefore, extended follow-up studies are needed for complete understanding of the long-term safety and the therapeutic efficacy of our allogeneic DC therapy.

The three transgenes were overexpressed to enhance antitumor immunity, based on published data suggesting their roles in activating T cells and B cells and promoting immune cell infiltration into tumors through chemotactic signaling. While this strategy aims to boost immune activation, we acknowledge that CD40L overexpression may carry a risk of systemic inflammation or autoimmunity. We have not observed any off-target side effects within our preclinical studies. Additional studies are needed to fully understand the safety profile (both in preclinical models and in patients), which includes local injection site reactions or transient flu-like symptoms, as observed with similar cell-based immunotherapies. Moreover, the advantage of utilizing allogeneic DCs over autologous DCs is that they do not have sustained survival once injected into the patient and are less likely to cause autoimmunity.

Our data demonstrated that treatment with engineered DCs induced an increase in the population of CD4+ T cells, CD8+ T cells, and NK cells and their ability to express IFN-γ and cytotoxic cell killing abilities. The increase in CD4^+^ T cells, CD8^+^ T cells, and NK cells in the spleen and peripheral blood indicates an effective systemic immune activation caused by engineered DCs. This enhanced immune response could play a key role in the regression of pancreatic tumors in mice treated with antigen-pulsed engineered DCs. While we agree that initial T cell priming likely occurs in tumor-draining lymph nodes (TDLNs), our analysis focused on the spleen and PBMCs as key sites of systemic immunity [68]. The elevated levels of CD8^+^ T cells in these compartments suggest successful priming and expansion, supporting the potential of our approach to target disseminated tumors. Importantly, ex vivo activation of splenocytes, PBMCs, and BM cells revealed enhanced cytotoxicity and IFN-γ production exclusively in cells from mice treated with engineered DCs. These functional responses underscore the capacity of this new therapy to elicit robust systemic immunity. Although lymph node responses were not directly assessed, our data strongly support the systemic immunostimulatory effects of the engineered DC therapy. We acknowledge the limitation of the small sample size in the animal study; however, we observe the impact of the engineered DCs on initiating an immune response leading to significant reductions in pancreatic tumors. Larger-scale studies are planned to confirm and extend these findings with appropriate statistical power in additional solid tumor models.

When assessing tumor infiltration in engineered DC-treated mice, we observe a significant increase in the numbers of CD4+ and CD8+ T cells in the tumors as compared to control treatments. These results indicated that treatment with the engineered DCs is effective not only in increasing the immune cells and their function, but more importantly they are effective in mobilizing T cells to the TME where they are needed to eliminate tumors. Studies have shown that replenishing deficient APC populations, using DC vaccines loaded with antigens from tumor lysates, can promote the transformation of a “cold” tumor into a “hot” tumor by inducing the activation and infiltration of CD8+ T lymphocytes to the TME [47]. In our study, this process was compounded with the genetic modification of immune-boosting genes (CD93, CD40L, and CXCL13), confirming a comprehensive and multi-faceted approach to overcome immune suppression and enhance immune activation. Myeloid-derived suppressor cells (MDSCs) and T regulatory cells (Tregs) are known to play an immunosuppressive role hindering overall therapeutic efficacy [69]. However, using engineered allogeneic DC therapy, we did not see such immune suppression. Further studies are needed to completely understand the relationship between engineered DC therapy and associated alterations in the tumor microenvironment.

MP2 cells express several well-characterized tumor-associated antigens commonly present in a wide subset of PCs, supporting their use as a representative antigen source. Nonetheless, we recognize that PC is highly heterogeneous and complex, with significant variations in antigen expression between patients and tumor subtypes. This heterogeneity poses challenges for antigen selection and underscores the need for personalized or multi-antigen approaches in future studies. Additionally, we acknowledge that the current study’s focus on MP2 lysates limits the breadth of applicability, as we have not yet evaluated our platform against other PC models or different tumor types. Expanding this research to include diverse tumor models will be crucial to fully assess the generalizability and translational potential of our approach, and we plan to address this in subsequent investigations.

Due to the antigen-presenting nature of DCs [70], data from multiple clinical trials have shown that DC vaccines are safer to administer and have a reductive effect on tumors. However, only one DC vaccine has so far been approved by the FDA (PROVENGE^®^ (Sipuleucel-T)) and the majority of clinical trials involving DC vaccines have shown limited therapeutic success [47,71,72]. The majority of DC vaccines being tested at the clinical stage are autologous DC vaccines comprising several technical challenges such as reproducibility, scalability, and host immune suppression that are difficult to overcome by using the autologous DC platforms [73]. The proposed off-the-shelf gene-modified allogeneic DC therapy offers a scalable, and standardized platform targeting pancreatic tumors with enhanced efficacy and significantly reduced toxicity compared to conventional treatments like chemotherapy and radiation therapy. The engineered DC therapeutic platform aims to lessen these burdens by reducing hospital stays, minimizing side effects, and improving patients’ overall quality of life and treatment accessibility. By using CD34+ HSCs, this innovative platform lowers manufacturing and delivery expenses overall by 20–30%, compared to conventional therapies (that include MoDCs). The engineered allogeneic DC platform presents an alternative option that could overcome the hurdles posed by autologous vaccines for the treatment of solid tumors [74].

## 5. Conclusions

Our data shows the therapeutic efficacy of engineered DCs in preclinical humanized mouse models bearing pancreatic tumors. This approach could offer an off-the-shelf treatment for solid tumors in the future. This allogeneic DC platform could overcome cancer immune suppression, induce long-lasting memory for surveillance, and inhibit tumor recurrence. Furthermore, our proposed engineered allogeneic DC platform can potentially be combined with other therapeutic modalities such as checkpoint inhibitor combinations, targeted radioligand therapies, and standard-of-care chemotherapy as a therapeutic modality for the treatment of hard-to-treat cancers [47,75].

## Figures and Tables

**Figure 4 vaccines-13-00749-f004:**
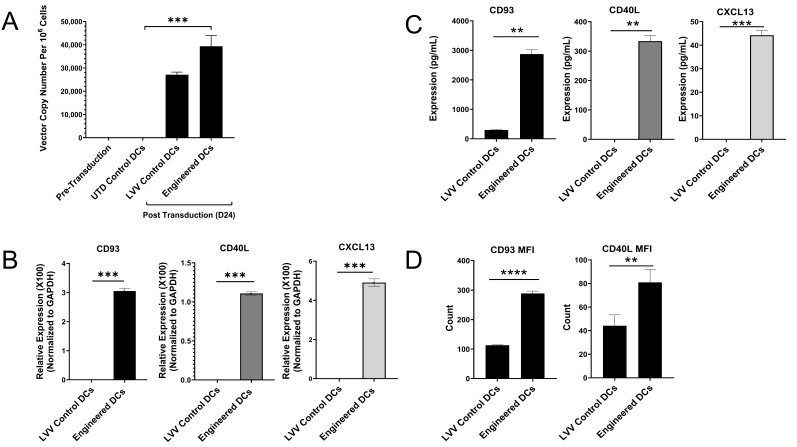
Expression of CD93, CD40L, and CXCL13 in engineered DCs. (**A**) Vector copy number analysis (*n* = 3, *** *p* value < 0.001 for engineered vs. UTD). (**B**) RT-qPCR data comparing the mRNA expression of transgenes in DC groups by normalizing to GAPDH (*n* = 3, *** *p* < 0.001). (**C**) The protein expression of transgenes CD93, CD40L, and CXCL13 following lentiviral transduction by ELISA (*n* = 3, CD93 and CD40L: ** *p* < 0.01, CXCL13: *** *p* < 0.001). (**D**) Surface expression analysis of CD93 and CD40L in engineered DCs by flow cytometry. Data shown as mean fluorescence intensity (MFI) ± SD, (*n* = 7, CD93: **** *p* < 0.0001, CD40L: *p* < 0.01).

**Figure 5 vaccines-13-00749-f005:**
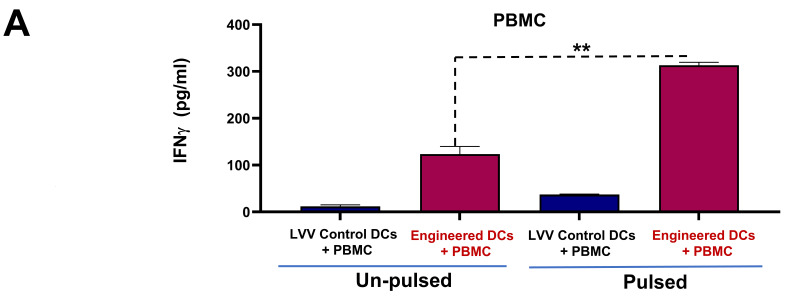
Engineered DCs induce T cell activation. (**A**) IFN-γ secretion in co-cultures of PBMCs with engineered DCs or LVV control DCs for 24 h. Data are presented as mean ± standard deviation (** *p*-value of <0.01). (**B**) Real-time monitoring of cytotoxicity using xCELLigence RTCA eSight system. Evaluated in a co-culture that includes pancreatic tumor cells (MP2) + DCs and CD8+ T cells. Data are presented as mean % of cytolysis for the high E/T (1.25:1) ratio over time until 40 h. (**C**) Representative images illustrating the morphological differences and MP2-GFP cell killing by T cells in different co-culture conditions. (**D**) Antigen-specific cytotoxicity of CD8^+^ T cells primed by tumor lysate-pulsed DCs. CD8^+^ T cells co-cultured with engineered DCs pulsed with pancreatic tumor lysate selectively killed pancreatic tumor cells but not unrelated hepatocellular carcinoma (SNU 423-GFP) cells.

**Figure 6 vaccines-13-00749-f006:**
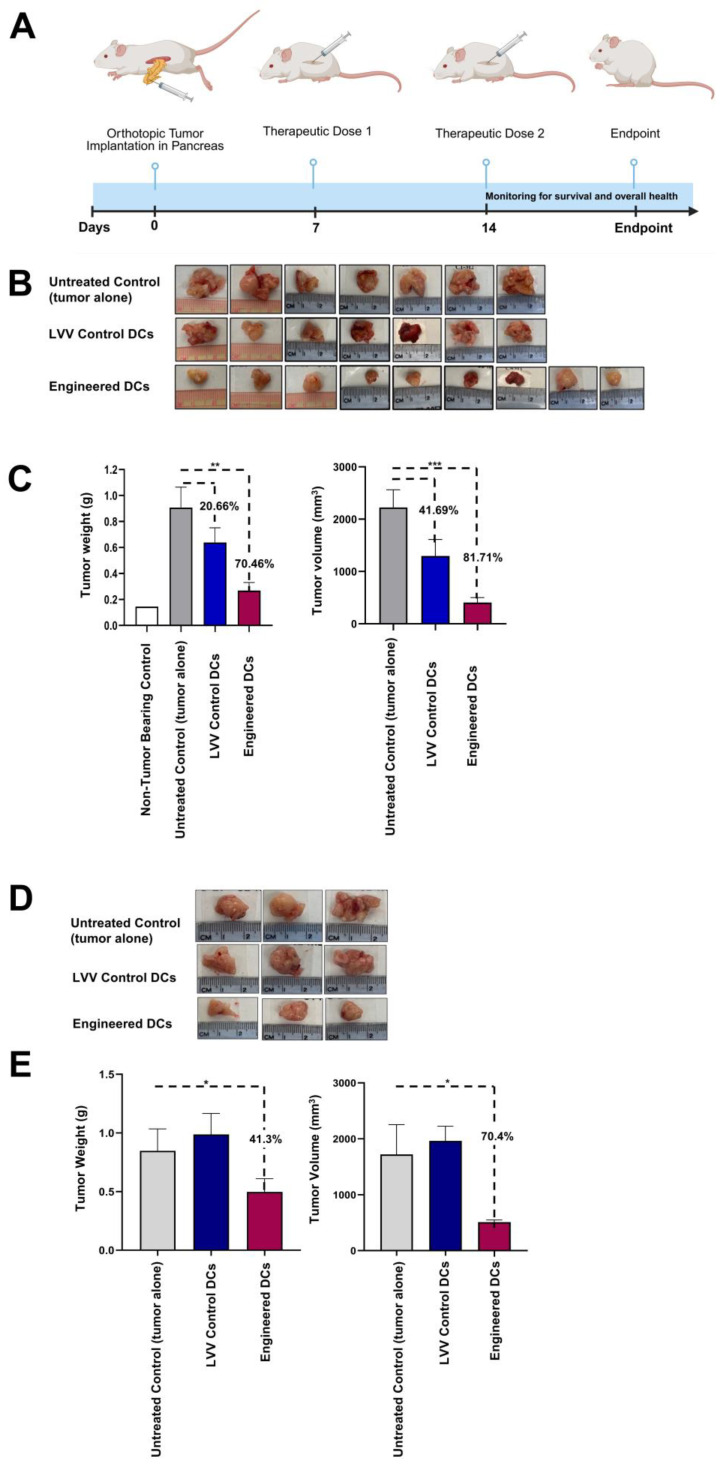
Pancreatic tumor regression by engineered DC treatment in humanized mouse models. (**A**) Schematic of the PC mouse model. (**B**) Representative tumor micrographs from each experimental group. (**C**) Graph representing tumor weight and volume data. Results are shown as mean ± standard deviation of tumor weight (grams) or tumor volume (mm^3^) from three independent experiments, *n* = 7 for control groups and *n* = 9 for experimental groups. Statistical significance: ** *p* = 0.001 for LVV control DCs vs. untreated tumor; *** *p* = 0.0001 for engineered DCs vs. untreated tumor. (**D**) Representative tumor micrographs from each experimental group. (**E**) Graph representing tumor weight and volume data. Data represents mean ± standard deviation of tumor weight (grams) or tumor volume (mm^3^) (*n* = 3 mice/group) (* *p* = 0.01 for engineered DCs vs. untreated tumor).

**Figure 7 vaccines-13-00749-f007:**
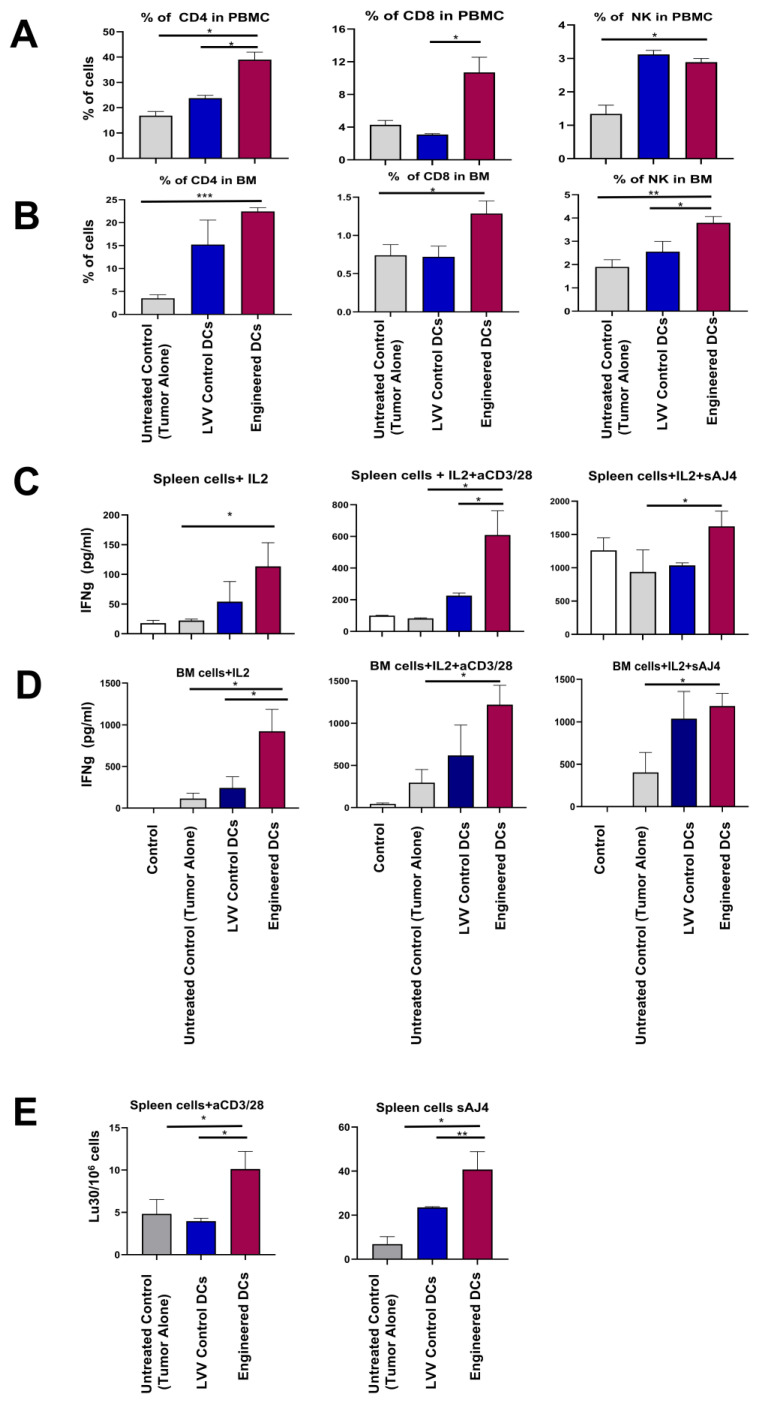
Engineered DCs enhance immune cell populations, IFN-γ secretion, and cytotoxic function in hu-BLT mice with pancreatic tumors. Flow cytometry analysis of PBMCs (**A**) and BM (**B**) from hu-BLT described in Figure 5. Statistical significance: * *p* < 0.05, ** *p* < 0.001 < *** *p* < 0.0010.001 and for engineered DCs vs. LVV control DCs or untreated controls. PBMCs (**C**) and BM (**D**) cells from each group were cultured for 5 days in the presence of IL-2, IL-2 + αCD3/28, or IL-2 + sAJ4. Supernatants were collected on day 5 and IFN-γ secretion was measured by ELISA. Data are presented as mean ± standard deviation of IFN-γ levels (pg/mL). Statistical sign * ficance: *p* < 0.05 for engineered DCs vs. LVV control DCs or untreated controls. (**E**) Splenocytes from each group were cultured for 5 days under the same conditions (IL-2, IL-2 + αCD3/28,and IL-2 + sAJ4) and then used as effector cells in a standard 4 h ^51^Cr release assay against OSCSCs. Lytic unit (LU) 30/10^6^ cells were calculated, representing the inverse number of splenocytes required to lyse 30% of OSCSCs × 100. Data is shown as mean ± standard deviation of LU 30/10^6^ cells. Statistical signi * ficance: *p* < 0.05 for engineered DCs vs. LVV control DCs or untreated controls.

**Figure 8 vaccines-13-00749-f008:**
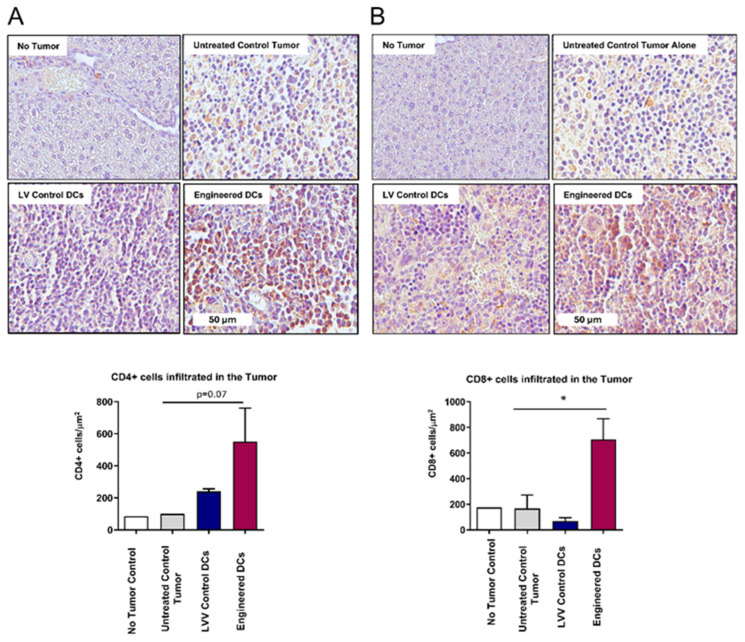
Engineered DCs enhance tumor infiltration of T cells in vivo. IHC analyses of pancreatic tumors isolated from hu-BLT mice for CD4+ T cells (**A**) and CD8+ T cells (**B**). Data are presented as mean ± standard deviation per 200 μm^2^ of tumor or normal tissue. Statistical significance: *p* < 0.05 for engineered DCs vs. LVV control DCs or untreated controls in CD8+ T cells. For CD4+ T cells, the difference was not statistically significant between the engineered DCs and control groups (*p* = 0.07).

## Data Availability

Data are contained within the article and Appendix A.

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
