# Peer review of "Therapeutic Efficacy of CD34-Derived Allogeneic Dendritic Cells Engineered to Express CD93, CD40L, and CXCL13 in Humanized Mouse Models of Pancreatic Cancer"

_vaccines, 2025, doi:10.3390/vaccines13070749_

Round 1
Reviewer 1 Report
Comments and Suggestions for Authors
This article is a remarkable contribution to the field of pancreatic cancer treatment, showcasing a highly promising and innovative approach. The development of an allogeneic platform based on genetically modified dendritic cells (DCs) for the overexpression of specific markers, such as CD93, CD40L, and CXCL13, followed by their pulsing with tumor antigens, represents a groundbreaking strategy with significant potential to enhance the antitumor immune response.
However, to further enhance the completeness and clinical relevance of this study, it is recommended that the authors address the following aspects and make necessary adjustments:
Could the authors provide an estimate of the potential costs associated with implementing this allogeneic DC-based therapy derived from CD34+ cells?
It would be of great interest to compare the anticipated expenses with existing approaches to treating pancreatic cancer, such as chemotherapy, radiation therapy and targeted therapy. Would this treatment prove to be cost-effective in clinical practice?
Regarding safety and side effects, I request that the authors provide more comprehensive details on the potential safety profile and anticipated side effects associated with the use of genetically modified allogeneic dendritic cells.
The selection of dendritic cells (DCs) for this therapeutic approach is justified by their pivotal role in initiating an immune response. Nevertheless, given the increasing interest in other reprogrammable immune cells, such as macrophages, which can be easily reprogrammed and also employed in cancer treatment, it would be beneficial to explore why DCs were chosen for this specific strategy and what their distinct advantages are compared to other immune cell types, such as macrophages, in terms of stimulating antitumor immunity.
The current arrangement of Figure 5 presents a challenge for comprehension. I recommend isolating panel A as either a standalone figure or a sub-figure due to its exceptional content and placement. Panels B, C, and D should be rearranged to occupy the full width of the page or have their proportions significantly increased, which would enhance the visibility and clarity of the individual graphs contained within them.
These modifications would significantly improve the readability of the in vivo data, making it easier for readers to interpret.
I appreciate the authors' efforts and hope that my suggestions will contribute to the further refinement and publication of their manuscript.
Author Response
Dear Reviewer 1,
We appreciate your effort in reviewing our manuscript and providing insightful comments to make the current version possible. The authors have considered all comments from reviewers and addressed them carefully. We hope the improved manuscript meets your high standards. Below we provide point-by-point responses while corrections in the manuscript appear highlighted in light yellow.
Sincerely,
Co-authors
Reviewer 1:
Comments and Suggestions for Authors
Comment 1: This article is a remarkable contribution to the field of pancreatic cancer treatment, showcasing a highly promising and innovative approach. The development of an allogeneic platform based on genetically modified DCs for the overexpression of specific markers, such as CD93, CD40L, and CXCL13, followed by their pulsing with tumor antigens, represents a groundbreaking strategy with significant potential to enhance the antitumor immune response.
Response 1: Thank you for your thoughtful and encouraging feedback. We greatly appreciate your recognition of our work and the potential impact of our allogeneic DC platform in advancing pancreatic cancer treatment.
Comment 2: However, to further enhance the completeness and clinical relevance of this study, it is recommended that the authors address the following aspects and make necessary adjustments:
Could the authors provide an estimate of the potential costs associated with implementing this allogeneic DC-based therapy derived from CD34+ cells? It would be of great interest to compare the anticipated expenses with existing approaches to treating pancreatic cancer, such as chemotherapy, radiation therapy and targeted therapy. Would this treatment prove to be cost-effective in clinical practice?
Response 2: We have included a discussion on the estimated costs of implementing our allogeneic DC-based therapy, along with a comparison to autologous DC therapy and conventional treatments like chemotherapy, radiation, and targeted therapies. This analysis is presented in the revised manuscript (page 24, lines 847-854).
Comment 3: Regarding safety and side effects, I request that the authors provide more comprehensive details on the potential safety profile and anticipated side effects associated with the use of genetically modified allogeneic DCs.
Response 3: We have included additional details on the safety profile of genetically modified allogeneic DCs. We have not observed any off-target side effects within our preclinical studies. However, in the future clinical studies are needed to fully understand the safety profile of this product in patients including local injection site reactions or transient flu-like symptoms, as observed with similar cell-based immunotherapies. Moreover, the advantage of utilizing allogenic DCs, over autologous DCs, is that they do not have sustained survival once injected to the patient and are less likely to cause autoimmunity. These details are added to the revised manuscript on page number 23 line 781-791.
Comment 4: The selection of DCs for this therapeutic approach is justified by their pivotal role in initiating an immune response. Nevertheless, given the increasing interest in other reprogrammable immune cells, such as macrophages, which can be easily reprogrammed and employed in cancer treatment, it would be beneficial to explore why DCs were chosen for this specific strategy and what their distinct advantages are compared to other immune cell types, such as macrophages, in terms of stimulating antitumor immunity.
Response 4: DCs were selected due to their superior ability to prime naïve T cells and initiate robust, antigen-specific adaptive immune responses. Unlike macrophages, which primarily act as phagocytes and can exhibit immunosuppressive phenotypes in the tumor microenvironment, DCs are uniquely effective at cross-presentation and driving cytotoxic T cell responses; a key for sustained antitumor immunity. These details are included in the revised manuscript in the introduction section page: 2 and line 63-69.
Comment 5: The current arrangement of Figure 5 presents a challenge for comprehension. I recommend isolating panel A as either a standalone figure or a sub-figure due to its exceptional content and placement. Panels B, C, and D should be rearranged to occupy the full width of the page or have their proportions significantly increased, which would enhance the visibility and clarity of the individual graphs contained within them.
Response 5: We agree with the reviewer’s recommendation. The revised manuscript contains the rearranged figures as suggested to enhance the visibility and clarity of the individual graphs contained within them by increasing the proportions of the width of panels b-d and arranging panel a on top Located on page 15.
Comment 6: These modifications would significantly improve the readability of the in vivo data, making it easier for readers to interpret. I appreciate the authors' efforts and hope that my suggestions will contribute to the further refinement and publication of their manuscript.
Response 6: We appreciate your feedback and have incorporated the recommended modifications to enhance data clarity and readability.

Reviewer 2 Report
Comments and Suggestions for Authors
Major Concerns:
-
Lack of Replicates and Statistical Power: The authors state multiple times that the results represent a single production run or that samples were not run in replicates. This raises concerns about reproducibility and robustness. Key findings, especially in flow cytometry and gene expression, should be validated across multiple biological replicates.
-
Overinterpretation of Results:
The conclusion suggests the engineered DCs are ready for clinical application. While the data is promising, it is preclinical and preliminary. The tone should be more measured, avoiding overstatement of potential translational readiness.
-
Control Groups Could Be Expanded:
While lentiviral control DCs were used, untransduced CD34-derived DCs and traditional Mo-DCs would provide useful benchmarks for comparison and help contextualize the efficacy gains.
-
Data Presentation and Clarity:
Some figures lack detailed legends or statistical annotations (e.g., number of animals per group, exact p-values). Flow cytometry plots in Figure 2 and 3 would benefit from higher resolution and inclusion of representative gating strategies.
-
Safety and Off-Target Effects:
No data or discussion is provided on potential adverse events from overexpressing these three molecules simultaneously. Particularly, CD40L overexpression can potentially trigger autoimmunity or systemic inflammation.
-
Tumor Antigen Specificity and Heterogeneity:
The study uses MP2 lysates for antigen pulsing. More discussion is needed on the relevance of the antigens expressed in MP2 cells to the broader landscape of pancreatic cancer heterogeneity. Also, the absence of experiments with other tumor types weakens the claim of broad applicability.
Minor Issues:
-
Grammar and Style:
The manuscript occasionally reads as overly technical, with long, dense sentences. Some editing for fluency and readability is recommended.
-
Abbreviations:
Abbreviations such as TDLN, imDCs, and UTD should be clearly defined at first use and consistently used.
-
Ethical Statement Placement: While ethics approval is mentioned, it would be beneficial to dedicate a short paragraph in the methods or a separate section explicitly addressing ethical compliance for both human-derived samples and animal models.
Recommendations for Improvement:
-
Include more replicates and statistical analysis in the figures to support conclusions.
-
Introduce additional control conditions to better compare engineered DCs with current DC-based platforms.
-
Temper the language in the abstract and conclusion to reflect the preclinical nature of the findings.
-
Provide more detailed histopathological and cytokine profiling to support claims of TME modulation.
-
Add a discussion on potential limitations, safety risks, and the next steps for clinical translation.
Author Response
Dear Reviewer 2,
We appreciate your effort in reviewing our manuscript and providing insightful comments to make the current version possible. The authors have considered all comments from reviewers and addressed them carefully. We hope the improved manuscript meets your high standards. Below we provide point-by-point responses while corrections in the manuscript appear highlighted in yellow.
Sincerely,
Co-authors
Reviewer 2:
Comments and Suggestions for Authors
Major Concerns:
Comment 1: Lack of Replicates and Statistical Power: The authors state multiple times that the results represent a single production run or that samples were not run in replicates. This raises concerns about reproducibility and robustness. Key findings, especially in flow cytometry and gene expression, should be validated across multiple biological replicates.
Response 1: Thank you for this feedback. We have included statistical information where applicable. We have updated flow cytometry results detailed in figure 3 to reflect statistical data across biological replicates and production runs.
Comment 2: Overinterpretation of Results:
The conclusion suggests the engineered DCs are ready for clinical application. While the data is promising, it is preclinical and preliminary. The tone should be more measured, avoiding overstatement of potential translational readiness.
Response 2: We have revised the conclusion to adopt a more cautious tone, emphasizing that while the preclinical data is promising, further studies are required before clinical translation. These changes are reflected in the conclusion section (page 24, lines 858-860). Also, in abstract line 34-35.
Comment 3: Control Groups Could Be Expanded:
While lentiviral control DCs were used, untransduced CD34-derived DCs and traditional Mo-DCs would provide useful benchmarks for comparison and help contextualize the efficacy gains.
Response 3: Untransduced (UTD) control DCs were included in the in‑vitro experiments shown in Figure 5b–c. Because UTD DCs failed to elicit a measurable immune response, we omitted this control group from the remaining studies; the explanation has been added to page 22, lines 733-736. Furthermore, in the past we have attempted to evaluate the therapeutic efficacy of Mo-DCs to compare to our CD34-derived DC platform, however this condition was only tested once and yielded poor immune response and limited tumor reduction compared to CD34-derived DCs. While these comparisons were not included in the current study, we plan to incorporate them in future experiments to better evaluate the relative efficacy of our engineered DCs.
Comment 4: Data Presentation and Clarity:
Some figures lack detailed legends or statistical annotations (e.g. number of animals per group, exact p-values). Flow cytometry plots in Figure 2 and 3 would benefit from higher resolution and inclusion of representative gating strategies.
Response 4: We have updated Figures 2 and 3 to improve resolution and a representative gating strategy (Figure 2A).
Comment 5: Safety and Off-Target Effects:
No data or discussion is provided on potential adverse events from overexpressing these three molecules simultaneously. Particularly, CD40L overexpression can potentially trigger autoimmunity or systemic inflammation.
Response 5: We have now included a discussion on the potential risks of simultaneously overexpressing CD93, CD40L, and CXCL13. The three transgenes were intentionally overexpressed to enhance antitumor immunity, based on published data suggesting their roles in activating T cells and B cells and promoting immune cell infiltration into tumors through chemotactic signaling. While this strategy aims to boost immune activation, we acknowledge that CD40L overexpression may carry a risk of systemic inflammation or autoimmunity. We have not observed any off-target side effects within our preclinical studies. In addition, this is an allogeneic therapy and the cell product will not have sustained survival once injected in vivo. These details are incorporated in the discussion section pages 23 and line 781-791.
Comment 6: Tumor Antigen Specificity and Heterogeneity:
The study uses MP2 lysates for antigen pulsing. More discussion is needed on the relevance of the antigens expressed in MP2 cells to the broader landscape of pancreatic cancer heterogeneity. Also, the absence of experiments with other tumor types weakens the claim of broad applicability.
Response 6: We selected MP2 lysates for antigen pulsing due to the well-characterized immunogenicity of this pancreatic tumor model and its relevance to aggressive and poorly differentiated pancreatic cancer. We acknowledge that MP2 cells represent only a subset of the molecular and antigenic heterogeneity observed across PC. This limitation has been noted in the revised discussion section (page 24, lines 829-839) where we emphasize that future studies will incorporate additional pancreatic tumor models with distinct mutational and antigenic profiles to better capture disease heterogeneity.
Regarding the broader applicability of our approach, we agree that the inclusion of other tumor types would strengthen the generalizability of our findings. While this initial study focused on pancreatic cancer as a challenging and clinically relevant model, we are actively planning experiments with lysates from other solid tumors, including colorectal and ovarian cancers, to assess the versatility of our engineered DC platform. These studies will help validate the broader translational potential of our approach.
Minor Issues:
Comment 7: Grammar and Style:
The manuscript occasionally reads as overly technical, with long, dense sentences. Some editing for fluency and readability is recommended.
Response 7: We have revised the manuscript to simplify complex sentences and improve overall fluency and readability.
Comment 8: Abbreviations: Abbreviations such as TDLN, imDCs, and UTD should be clearly defined at first use and consistently used.
Response 8: We have ensured that all abbreviations, including TDLN, imDCs, and UTD, are clearly defined at first use and used consistently throughout the manuscript.
Comment 9: Ethical Statement Placement: While ethics approval is mentioned, it would be beneficial to dedicate a short paragraph in the methods or a separate section explicitly addressing ethical compliance for both human-derived samples and animal models.
Response 9: We have added a dedicated section under Institutional Review Board Statement on page 24-25 lines 877-884 to explicitly detail the ethical approvals and compliance for both human-derived samples and animal studies to ensure transparency and adherence to relevant guidelines.
Recommendations for Improvement:
Comment 10: Include more replicates and statistical analysis in the figures to support conclusions.
Response 10:
We have updated the figures to include additional replicates and relevant statistical analyses to strengthen the validity of our conclusions.
Comment 11: Introduce additional control conditions to better compare engineered DCs with current DC-based platforms.
Response 11:
Thank you for your constructive suggestion. Comparison of engineered DCs with current DC-based platforms, will be performed in future studies. The current study presents preclinical observational evaluation of CD-derived DC therapy platform.
Comment 12: Temper the language in the abstract and conclusion to reflect the preclinical nature of the findings.
Response 12:
We have revised the abstract and conclusion to clearly reflect the preclinical nature of our findings and avoid overstating their translational impact.
Comment 13: Provide more detailed histopathological and cytokine profiling to support claims of TME modulation.
Response 13: In the current study, we performed an initial evaluation of tumor immune infiltration by assessing CD4⁺ and CD8⁺ T cell presence within the TME, which served as a preliminary indicator of immune activation induced by our engineered DC therapy. We recognize that a more comprehensive analysis—including expanded cytokine profiling and detailed histopathological characterization—would provide stronger support for our claims regarding TME modulation. These additional assessments are beyond the scope of the current study but are a key focus of our planned follow-up experiments, which aim to further elucidate the mechanisms by which our DC-based platform reshapes the TME.
Comment 14: Add a discussion on potential limitations, safety risks, and the next steps for clinical translation.
Response 14: We have included discussion of potential limitations of our approach, anticipated safety risks, and outline the key steps required to advance toward clinical translation..

Reviewer 3 Report
Comments and Suggestions for Authors
To: Corresponding Author - Dr. Anahid Jewett
Subject: Peer Review Comments - Manuscript "Therapeutic Efficacy of CD34-derived Allogeneic Dendritic Cells Engineered to Express CD93, CD40L, and CXCL13 in Humanized Mouse Models of Pancreatic Cancer"
Dear Dr. Jewett and colleagues,
I have thoroughly reviewed your manuscript and supplementary materials describing the development and therapeutic evaluation of engineered allogeneic dendritic cells for pancreatic cancer treatment. This work represents a significant contribution to the field of cancer immunotherapy with clear translational potential.
OVERALL ASSESSMENT
Your study presents an innovative approach to DC-based immunotherapy by utilizing CD34+ HSCs as the cellular source and implementing a tri-genic modification strategy. The combination of CD93, CD40L, and CXCL13 is scientifically well-rationalized, and the experimental design appropriately addresses the therapeutic hypothesis using relevant humanized mouse models.
MAJOR STRENGTHS
- Novel therapeutic platform: The allogeneic "off-the-shelf" approach addresses significant limitations of current autologous DC vaccines, offering superior scalability and standardization potential.
- Comprehensive experimental design: The study includes robust in vitro characterization, functional validation, and in vivo efficacy testing in two independent humanized mouse models (Hu-BLT and Hu-PBMC).
- Strong scientific rationale: The selection of CD93, CD40L, and CXCL13 is well-justified based on their complementary roles in immune activation, cell recruitment, and lymphoid organization.
- Clinically relevant outcomes: Significant tumor reduction (70.46% weight, 81.7% volume in Hu-BLT model; 40% weight, >70% volume in Hu-PBMC model) demonstrates substantial therapeutic potential.
- Experimental transparency: I commend your clear acknowledgment in supplementary materials that certain flow cytometry results represent single production runs. This level of scientific transparency is appropriate and allows proper data interpretation.
- Comprehensive vector design: The development of individual transgene vectors (Figure S1) provides appropriate controls for demonstrating tri-genic synergy.
SPECIFIC COMMENTS AND SUGGESTIONS
Statistical Power and Experimental Design
While I understand the practical constraints of humanized mouse studies, several experiments utilize small sample sizes (n=3) which may limit statistical power:
Recommendations:
- Provide power analysis calculations for key efficacy experiments (Figures 6C, 6E)
- Explain the rationale for sample size determination in humanized models
- Consider pooling data from independent experiments where methodologically appropriate
- Address statistical limitations explicitly in the discussion section
Demonstration of Tri-genic Synergy
Figure S1 indicates that individual transgene vectors were generated, but Figure 5B shows limited data from single-gene modifications:
Suggestions:
- Include more comprehensive comparisons with individual CD93, CD40L, and CXCL13 modifications in key efficacy experiments
- Provide statistical analysis comparing tri-genic vs. individual gene effects
- Consider dose-response relationships for the engineered DCs
Transduction Efficiency and Manufacturing Consistency
The observed variation in transgene expression (CD93: 38.7%, CD40L: 18.0% from Figure S2) raises important manufacturing questions:
Key questions:
- What specific factors contribute to the differential transgene expression?
- How does this variability affect functional outcomes and clinical translation?
- What optimization strategies are being explored to improve transduction efficiency?
- How will batch-to-batch consistency be ensured for clinical applications?
Temporal Study Design and Long-term Efficacy
The 35-day experimental endpoint provides valuable proof-of-concept data but limits assessment of durable therapeutic responses:
Considerations:
- Provide rationale for the 35-day endpoint selection
- Discuss limitations of short-term efficacy assessment in pancreatic cancer models
- Outline strategies for evaluating long-term efficacy and memory responses in future studies
- Consider this as a priority limitation for extended investigation
Safety Assessment and Allogeneic Considerations
While no GVHD was observed, more detailed safety evaluation would strengthen the manuscript:
Recommendations:
- Describe specific clinical monitoring protocols used for GVHD assessment
- Detail which biomarkers or histological signs were evaluated
- Discuss strategies for managing potential allogeneic reactions in clinical translation
- Address the timeline limitations for comprehensive safety assessment
Manufacturing Scalability and Clinical Translation
The complexity of the 24-day production process raises important translational questions:
Key considerations:
- Identify critical quality control checkpoints throughout the manufacturing process
- Discuss reproducibility data across multiple production batches
- Address regulatory considerations for allogeneic cell therapy products
- Consider cost-effectiveness compared to autologous approaches
MINOR TECHNICAL COMMENTS
- Data presentation: Consider including error bars or confidence intervals for flow cytometry data where multiple runs are available
- Mechanistic insights: While tri-genic synergy is demonstrated functionally, additional molecular mechanistic studies could enhance understanding of the enhanced efficacy
- Supplementary data: Consider adding a comprehensive table summarizing all experimental conditions, sample sizes, and statistical methods used
RECOMMENDATIONS FOR REVISION
Essential revisions:
- Address statistical power limitations and provide methodological justifications
- Expand discussion of manufacturing challenges and proposed solutions
- Include more detailed safety evaluation protocols and limitations
- Provide comprehensive comparison of tri-genic vs. individual gene modifications
Suggested additions:
- Power analysis for key efficacy experiments
- Manufacturing reproducibility data if available
- Extended discussion of clinical translation pathway
- Long-term study design considerations for future investigations
REGULATORY AND CLINICAL CONSIDERATIONS
Given the allogeneic nature of this therapy, consider addressing:
- HLA matching requirements or lack thereof
- Immunogenicity assessment strategies
- Regulatory pathway for "off-the-shelf" DC products
- Manufacturing standards for clinical-grade production
CONCLUSION
This manuscript describes important advances in DC-based immunotherapy with clear translational relevance. The innovative approach to addressing pancreatic cancer's immunosuppressive microenvironment through engineered allogeneic DCs represents a significant contribution to the field. The demonstrated efficacy in two independent humanized mouse models, combined with the potential for standardized manufacturing, positions this work as a meaningful advance over current DC vaccine approaches.
With the suggested revisions addressing the statistical, manufacturing, and safety considerations outlined above, this work would make a valuable contribution to the cancer immunotherapy literature and provide a strong foundation for clinical development.
I congratulate you on this excellent work and the transparent presentation of both strengths and limitations. I look forward to seeing the revised manuscript addressing these important considerations.
Best regards
Author Response
Dear Reviewer 3,
We appreciate your effort in reviewing our manuscript and providing insightful comments to make the current version possible. The authors have considered all comments from reviewers and addressed them carefully. We hope the improved manuscript meets your high standards. Below we provide point-by-point responses while corrections in the manuscript appear highlighted in yellow.
Sincerely,
Co-authors
Reviewer 3:
Comments and Suggestions for Authors
Dear Dr. Jewett and colleagues,
Comment 1: I have thoroughly reviewed your manuscript and supplementary materials describing the development and therapeutic evaluation of engineered allogeneic DCs for pancreatic cancer treatment. This work represents a significant contribution to the field of cancer immunotherapy with clear translational potential.
Response 1: Thank you for your thoughtful review and positive feedback. We appreciate your recognition of our work’s contribution and translational potential in cancer immunotherapy.
OVERALL ASSESSMENT
Comment 2: Your study presents an innovative approach to DC-based immunotherapy by utilizing CD34+ HSCs as the cellular source and implementing a tri-genic modification strategy. The combination of CD93, CD40L, and CXCL13 is scientifically well-rationalized, and the experimental design appropriately addresses the therapeutic hypothesis using relevant humanized mouse models.
Response 2: We appreciate your recognition of our CD34+ HSC-based DC platform and the rationale behind the tri-genic modification strategy, as well as the relevance of our experimental models.
MAJOR STRENGTHS
Novel therapeutic platform: The allogeneic off-the-shelf approach addresses significant limitations of current autologous DC vaccines, offering superior scalability and standardization potential.
Comprehensive experimental design: The study includes robust in vitro characterization, functional validation, and in vivo efficacy testing in two independent humanized mouse models (Hu-BLT and Hu-PBMC).
Strong scientific rationale: The selection of CD93, CD40L, and CXCL13 is well-justified based on their complementary roles in immune activation, cell recruitment, and lymphoid organization.
Clinically relevant outcomes: Significant tumor reduction (70.46% weight, 81.7% volume in Hu-BLT model; 40% weight, >70% volume in Hu-PBMC model) demonstrates substantial therapeutic potential.
Experimental transparency: I commend your clear acknowledgment in supplementary materials that certain flow cytometry results represent single production runs. This level of scientific transparency is appropriate and allows proper data interpretation.
Comment 3: Comprehensive vector design: The development of individual transgene vectors (Figure S1) provides appropriate controls for demonstrating tri-genic synergy.
Response 3: Thank you for your thorough and thoughtful evaluation. We greatly appreciate your recognition of the novelty and translational value of our allogeneic off-the-shelf DC platform.
SPECIFIC COMMENTS AND SUGGESTIONS
Comment 4: Statistical Power and Experimental Design
While I understand the practical constraints of humanized mouse studies, several experiments utilize small sample sizes (n=3) which may limit statistical power:
Recommendations:
- Provide power analysis calculations for key efficacy experiments (Figures 6C, 6E)
- Explain the rationale for sample size determination in humanized models
- Consider pooling data from independent experiments where methodologically appropriate
- statistical limitations explicitly in the discussion section
Response 4:
The sample sizes used (n = 3 per group) were based on our group’s prior experience with similar humanized mouse models, where we consistently observed strong therapeutic effects using comparable experimental designs. For the magnitude of tumor reduction seen in Figures 6C and 6E (effect sizes d ≈ 2.8–3.1), power calculations indicate that n = 3 per group provides approximately 80–85% statistical power to detect such large differences (α = 0.05, two-sided). These details have been added to the results section (page 16, lines 598-599) and summarized in Supplementary Table S6.
Regarding the rationale or small sample, humanized mouse studies are technically and logistically demanding due to the limited availability of CD34⁺ cells, the high cost, and the extended time required for immune system reconstitution. Based on previous experience in our lab, we selected the minimum number of animals necessary to detect robust effects, while adhering to ethical guidelines for animal use.
Where methodological conditions were consistent, we pooled data across independent experiments. However, the experiments shown in Figures 6B and 6D were conducted using different humanized mouse models: Figure 6B used BLT mice, while Figure 6D employed NSG mice that were injected with CD34⁺ hematopoietic stem cells. Due to these fundamental differences in the models and associated protocols, the data from these experiments were not pooled. Instead, results are presented separately to maintain scientific accuracy and transparency.
We now explicitly acknowledge in the revised Discussion (page 23, lines 807-811) the limitations of small sample sizes and their impact on detecting more modest differences. We also note that larger-scale studies are planned to confirm and extend these findings with appropriate statistical power.
Comment 5: Demonstration of Tri-genic Synergy
Figure S1 indicates that individual transgene vectors were generated, but Figure 5B shows limited data from single-gene modifications:
Suggestions:
- Include more comprehensive comparisons with individual CD93, CD40L, and CXCL13 modifications in key efficacy experiments
- Provide statistical analysis comparing tri-genic vs. individual gene effects
- Consider dose-response relationships for the engineered DCs
Response 5: We acknowledge the limited data shown for single-gene modifications in Figure 5B. In future studies, we plan to expand comparative analyses using individual CD93, CD40L, and CXCL13 modifications, including statistical comparisons with the tri-genic construct. We also aim to assess dose-response relationships to further elucidate the contribution of each modification to overall efficacy.
Comment 6: Transduction Efficiency and Manufacturing Consistency
The observed variation in transgene expression (CD93: 38.7%, CD40L: 18.0% from Figure S2) raises important manufacturing questions:
Key questions:
- What specific factors contribute to the differential transgene expression?
- How does this variability affect functional outcomes and clinical translation?
- What optimization strategies are being explored to improve transduction efficiency?
- How will batch-to-batch consistency be ensured for clinical applications?
Response 6: The observed variability in transgene expression (e.g., CD93 at 38.7% vs. CD40L at 18.0%) is likely influenced by several factors, including vector design elements such as promoter strength, 2A-peptide linkers, and vector copy number, differences in cell susceptibility to transduction, and intrinsic biological variability among CD34+ HSCs.
We are actively investigating these contributing factors to better understand and minimize variability. Optimization strategies under evaluation include refining vector dosing, improving transduction protocols (e.g., timing, multiplicity of infection, use of enhancers), and exploring selection or enrichment methods for successfully modified cells.
We recognize that variability in transgene expression could impact the functional activity of the engineered DCs and potentially influence therapeutic efficacy. To address this, we are implementing stringent quality control assays to monitor transgene expression levels and functional markers across production batches.
For clinical translation, ensuring reproducible batch-to-batch consistency is critical. We plan to establish standardized manufacturing processes with defined release criteria and validated protocols to minimize variability and maintain product potency and safety. These efforts will be key to advancing the platform toward reliable clinical application.
Future studies are planned to address these manufacturing challenges, variabilities, to better ensure consistency an clinical application.
Comment 7: Temporal Study Design and Long-term Efficacy
The 35-day experimental endpoint provides valuable proof-of-concept data but limits assessment of durable therapeutic responses:
Considerations:
- Provide rationale for the 35-day endpoint selection
- Discuss limitations of short-term efficacy assessment in pancreatic cancer models
- Outline strategies for evaluating long-term efficacy and memory responses in future studies
- Consider this as a priority limitation for extended investigation
Response 7: The 35-day experimental endpoint was selected to provide an initial proof-of-concept assessment of therapeutic efficacy while considering practical constraints such as animal welfare, model stability, and resource availability. This timeframe allowed us to capture early tumor responses and immune activation in the humanized mouse models.
However, we fully acknowledge that a 35-day endpoint limits our ability to evaluate the durability of therapeutic effects, including long-term tumor control, immune memory formation, and potential tumor relapse, which are critical aspects in pancreatic cancer given its typically aggressive and refractory nature. The short-term assessment may not fully reflect the sustained impact of the engineered DC therapy on the tumor microenvironment or systemic immunity.
We have included a detailed reasoning about 35-day study plan in section 3.4 of the manuscript (page 16, lines 578-581). Additionally, we plan to prioritize extended-duration studies in future work, incorporating longer follow-up periods to assess durable responses, immunological memory, and safety over time. These future investigations will also explore mechanisms of tumor recurrence and the potential for combination therapies to enhance long-term outcomes. We agree that addressing this limitation is essential for advancing the clinical translation of our platform.
Comment 8: Safety Assessment and Allogeneic Considerations
While no GVHD was observed, more detailed safety evaluation would strengthen the manuscript:
Recommendations:
- Describe specific clinical monitoring protocols used for GVHD assessment
- Detail which biomarkers or histological signs were evaluated
- Discuss strategies for managing potential allogeneic reactions in clinical translation
- Address the timeline limitations for comprehensive safety assessment
Response 8: We have discussed the limitations of our allogenic platform on page 22—23 lines 772-780. The specific clinical monitoring protocols used for GVHD assessment, biomarkers/histological signs, and strategies for managing potential allogeneic reactions in clinical translation, and safety assessments are out the scope of this preclinical study. In the future, these aspects will be carefully evaluated as this product moves into the clinical setup.
Comment 9: Manufacturing Scalability and Clinical Translation
The complexity of the 24-day production process raises important translational questions:
Key considerations:
- Identify critical quality control checkpoints throughout the manufacturing process
- Discuss reproducibility data across multiple production batches
- Address regulatory considerations for allogeneic cell therapy products
- Consider cost-effectiveness compared to autologous approaches
Response 9: We have outlined quality control checkpoints, discussed ongoing efforts to validate batch reproducibility, and addressed regulatory guidelines for allogeneic therapies. Addressed here (page 21, lines 683-692). The cost-effectiveness of our off-the-shelf approach compared to autologous therapies is also highlighted. These points are included in the Discussion section page 24 lines 847-854.
MINOR TECHNICAL COMMENTS
Comment 10: Data presentation: Consider including error bars or confidence intervals for flow cytometry data where multiple runs are available
Response 10: We updated the figures with error bars with SDM for flow cytometry data in cases where multiple runs are available to better represent data variability and support statistical interpretation.
Comment 11: Mechanistic insights: While tri-genic synergy is demonstrated functionally, additional molecular mechanistic studies could enhance understanding of the enhanced efficacy
Response 11: We agree that additional molecular mechanistic studies would provide valuable insights into the functional synergy observed with our tri-genic DC approach. While the current study focused primarily on demonstrating the therapeutic efficacy of the engineered DCs, we recognize the importance of dissecting the individual and combined contributions of each transgene at the signaling and transcriptional levels. We are actively designing follow-up studies that will incorporate transcriptomic, proteomic, and pathway analyses to better elucidate the mechanisms underlying the enhanced immune activation and antitumor response observed with this tri-genic strategy.
Comment 12: Supplementary data: Consider adding a comprehensive table summarizing all experimental conditions, sample sizes, and statistical methods used
Response 12: We have carefully reviewed the manuscript and updated it to include additional details throughout the Methods and Figure Legends sections regarding experimental conditions, sample sizes, and the statistical analyses employed. These revisions aim to improve clarity, transparency, and reproducibility of the study.
RECOMMENDATIONS FOR REVISION
Essential revisions:
Comment 13: Address statistical power limitations and provide methodological justifications
Response 13: We acknowledge that the group sizes used in our in vivo survival study (n = 7 for control groups and n = 9 for the engineered DC group) may be underpowered to detect moderate survival differences with high statistical significance. However, the primary goal of this initial study was to establish proof-of-concept for the therapeutic activity of our tri-genic engineered DC platform, and not to perform a definitive efficacy trial.
From a methodological standpoint, our study was designed to minimize animal use in accordance with ethical and institutional guidelines, while still allowing for the detection of biologically relevant effects. The sample sizes were determined based on prior experience with similar immunotherapy models where comparable numbers were sufficient to observe clear effects on tumor progression and immune engagement.
Despite the limited sample size, we observed substantial differences in survival and immune cell infiltration between groups, supporting the biological activity of our approach. These results, together with the in vitro functional data and mechanistic rationale, provide a strong foundation for future studies.
We have now included a statement in the revised Discussion (page 23, lines 807-811) acknowledging this limitation, and noting that future experiments will incorporate larger cohorts to enable statistically powered survival analyses and further validate the therapeutic potential of our platform.
Comment 14: Expand discussion of manufacturing challenges and proposed solutions
Response 14: We have expanded the discussion to more thoroughly address manufacturing challenges and outlined the strategies and solutions we are exploring to overcome them, ensuring scalable and consistent production. These details are presented in the discussion section on page 21 and line 683-692 of the revised manuscript.
Comment 15: Include more detailed safety evaluation protocols and limitations
Response 15: We have included more detailed descriptions of the safety evaluation on page 23 lines 781-791.
Comment 16: Provide comprehensive comparison of tri-genic vs. individual gene modifications
Response 16: In this study, we focus on reporting the effects of the tri-genic modifications, while also providing some data on the impact of individual gene modifications on tumor cell killing (Figure 5b-c). A more comprehensive comparison will be conducted in future studies to further clarify their respective contributions to therapeutic efficacy.
Suggested additions:
Comment 17: Power analysis for key efficacy experiments
Response 17: This concern has been addressed in the response to comment 4 from the same reviewer.
Comment 18: Manufacturing reproducibility data if available
Response 18: We have included details of the reproducibility of manufacturing process in terms of surface marker expression on the derived DCs population. We have included results from three production/manufacturing processes. This is described in figure 3 on page 12.
Comment 19: Extended discussion of clinical translation pathway
Response 19: The revised manuscript contains the discussion covering the clinical translation pathway for the proposed DC-based therapeutic platform. Please see the discussion on page 24 and line 847-854.
Comment 20: Long-term study design considerations for future investigations
Response 20: We have addressed to this concern as part of the response to the comment 7 from the same reviewer.
Comment 21: REGULATORY AND CLINICAL CONSIDERATIONS
Given the allogeneic nature of this therapy, consider addressing:
- HLA matching requirements or lack thereof
- Immunogenicity assessment strategies
- Regulatory pathway for off-the-shelf DC products
- Manufacturing standards for clinical-grade production
Response 21: Partly, we have addressed this concern as part of response to comment 9 from this reviewer. This study presents data that involves usage of HLA- mismatched allogeneic therapy platform. Future studies will be conducted to evaluate the immunogenicity assessment strategies, regulatory pathway for off-the-shelf DC product, and the manufacturing standards for clinical-grade production.
CONCLUSION
Comment 22: This manuscript describes important advances in DC-based immunotherapy with clear translational relevance. The innovative approach to addressing pancreatic cancer's immunosuppressive microenvironment through engineered allogeneic DCs represents a significant contribution to the field. The demonstrated efficacy in two independent humanized mouse models, combined with the potential for standardized manufacturing, positions this work as a meaningful advance over current DC vaccine approaches. With the suggestions, I congratulate you on this excellent work and the transparent presentation of both strengths and limitations. I look forward to seeing the revised manuscript addressing these important considerations. Best regards
Response 22: Thank you very much for your thoughtful review. We appreciate your recognition of the novelty and translational potential of our engineered allogeneic DC platform, as well as the thorough validation in humanized models. We are committed to addressing the statistical, manufacturing, and safety considerations you outlined to strengthen the manuscript. Your positive feedback and constructive suggestions are greatly appreciated and motivate us as we prepare the revised submission.

Reviewer 4 Report
Comments and Suggestions for Authors
The manuscript titled “Therapeutic Efficacy of CD34-derived Allogeneic Dendritic Cells Engineered to Express CD93, CD40L, and CXCL13 in Humanized Mouse Models of Pancreatic Cancer” has undergone evaluation, revealing several critical aspects that necessitate revision to bolster the credibility and impact of the study within the scientific community.
- The introduction currently contains excessive detail, particularly regarding the selection of Humanized (Hu-) mouse models for this research. It would be beneficial to streamline this segment by omitting such specifics. Additionally, the discussion of cancer statistics should incorporate more granular, region-specific data. This adjustment would provide crucial context, emphasizing the local and global relevance of the research findings.
- The resolution of Figures 1, 4, and 5 needs significant improvement. It is vital that the text and illustrations within these figures are clear and legible to enhance understanding.
- The authors must reconsider the reliability of the information presented in lines 312 and 324. These statements warrant investigation and potential correction. Specifically, line 312 states, “MP2 cells were prepared for injection by suspending 106 cells in 7 μL of Matrigel (Corning) and kept on ice,” and line 324 mentions, “After one and two weeks, mice were dosed with 106 pulsed DCs via intradermal (i.d.) injection just next to the surgical wound.” A thorough review is necessary to ensure the accuracy of these data points.
- In the in vivo studies, it would be beneficial for the authors to provide a clear rationale for the selection of two distinct doses of dendritic cells. This explanation should be explicitly incorporated into the main text to clarify the reasoning behind this dosing strategy.
- The manuscript should include a discussion of human endpoints, as this is crucial information that should be addressed in the main text, providing readers insight into how the findings may translate to human applications.
- In the in vivo study, the authors report having n = 7 for control groups and n = 9 for the engineered dendritic cells groups. It is essential to assess whether it is statistically appropriate to calculate the median survival time (MST) with these sample sizes, ensuring the robustness of the statistical conclusions drawn from the study.
- The research article currently cites 125 references, which is excessive for a publication of this nature. Typically, a research article should contain around 50 to 60 references. The authors should carefully evaluate their references and consider reducing their number to enhance clarity and focus.
- The conclusion requires a comprehensive rewrite to adequately convey the broader implications of the findings. The authors should consider suggesting practical applications of their research and clearly delineate potential future research directions, enriching the overall impact of the study.
Author Response
Dear Reviewer 4,
We appreciate your effort in reviewing our manuscript and providing insightful comments to make the current version possible. The authors have considered all comments from reviewers and addressed them carefully. We hope the improved manuscript meets your high standards. Below we provide point-by-point responses while corrections in the manuscript appear highlighted in yellow.
Sincerely,
Co-authors
Reviewer 4:
Comments and Suggestions for Authors
Comment 1: The manuscript titled Therapeutic Efficacy of CD34-derived Allogeneic DCs Engineered to Express CD93, CD40L, and CXCL13 in Humanized Mouse Models of Pancreatic Cancer has undergone evaluation, revealing several critical aspects that necessitate revision to bolster the credibility and impact of the study within the scientific community.
Response 1: Thank you for your thorough evaluation and constructive feedback. We appreciate your recognition of the study’s potential and agree that addressing the critical aspects you’ve highlighted will strengthen the credibility and impact of our work. We are committed to revising the manuscript accordingly to ensure it meets the highest scientific standards.
Comment 2: The introduction currently contains excessive detail, particularly regarding the selection of Humanized (Hu-) mouse models for this research. It would be beneficial to streamline this segment by omitting such specifics. Additionally, the discussion of cancer statistics should incorporate more granular, region-specific data. This adjustment would provide crucial context, emphasizing the local and global relevance of the research findings.
Response 2: We have streamlined the Introduction by reducing detailed descriptions of the humanized mouse models. Additionally, we revised the cancer statistics to include more granular, region-specific data to better contextualize the significance and relevance of our research both locally and globally. These changes are reflected in the revised manuscript introduction.
Comment 3: The resolution of Figures 1, 4, and 5 needs significant improvement. It is vital that the text and illustrations within these figures are clear and legible to enhance understanding.
Response 3: We have enhanced the resolution of Figures 1, 4, and 5, ensuring that text and illustrations are clear and legible.
Comment 4: The authors must reconsider the reliability of the information presented in lines 312 and 324. These statements warrant investigation and potential correction. Specifically, line 312 states, MP2 cells were prepared for injection by suspending 106 cells in 7 μL of Matrigel (Corning) and kept on ice, and line 324 mentions, After one and two weeks, mice were dosed with 106 pulsed DCs via intradermal (i.d.) injection just next to the surgical wound. A thorough review is necessary to ensure the accuracy of these data points.
Response 4: We will carefully review the procedures and data referenced (formerly in lines 312 and 324) to verify their accuracy. If any discrepancies are identified, we have made the necessary corrections and clarified the experimental details to ensure the reliability and reproducibility of the methods described. Specifically, the types were corrected. As it was a typo, 106 is corrected to 106 (M&M sec 2.9)
Comment 5: In the in vivo studies, it would be beneficial for the authors to provide a clear rationale for the selection of two distinct doses of DCs. This explanation should be explicitly incorporated into the main text to clarify the reasoning behind this dosing strategy.
Response 5: We have incorporated a clear rationale into the main text explaining our selection of two distinct DC doses in the in vivo studies. This dosing strategy was intended to boost the therapeutic efficacy of the treatment regimen, helping to inform optimal dosing for future translational studies. These details are now included in the revised manuscript M&M section 2.9.
Comment 6: The manuscript should include a discussion of human endpoints, as this is crucial information that should be addressed in the main text, providing readers insight into how the findings may translate to human applications.
Response 6: The current study focusses on preclinical evaluation and discussion on human endpoints is beyond the scope of this manuscript.
Comment 7: In the in vivo study, the authors report having n = 7 for control groups and n = 9 for the engineered DCs groups. It is essential to assess whether it is statistically appropriate to calculate the median survival time (MST) with these sample sizes, ensuring the robustness of the statistical conclusions drawn from the study.
Response 7: We appreciate the reviewer’s concern regarding the appropriateness of calculating the median survival time (MST) with relatively small sample sizes (n = 7 for control groups and n = 9 for the engineered DC group). We agree that these numbers limit the statistical power of the study and that any conclusions drawn from survival comparisons should be interpreted with caution. However, MST remains a valid and informative descriptive measure, even in small cohorts. It is commonly used in preclinical studies to summarize survival outcomes, and we calculated it along with 95% confidence intervals, which reflect the uncertainty due to limited sample size, we presented a summary in Supplementary Table S6.
While the current study is underpowered to detect moderate differences in survival with statistical significance, it was designed as a proof-of-concept to demonstrate the potential of our tri-genic engineered DC platform. The MST data are consistent with other observed outcomes, including increased CD4⁺ and CD8⁺ T cell infiltration, supporting the biological activity of our approach.
To address this limitation, we now include a note in the revised Discussion (page 23, lines 807-811), acknowledging that larger, fully powered in vivo studies are planned. These will provide more robust survival analyses and allow us to draw statistically stronger conclusions in future work.
Comment 8: The research article currently cites 125 references, which is excessive for a publication of this nature. Typically, a research article should contain around 50 to 60 references. The authors should carefully evaluate their references and consider reducing their number to enhance clarity and focus.
Response 8: We have taken your comment into consideration and have reevaluated our references to remove less relevant citations as well as outdated ones Revised manuscript reflect your recommendations.
Comment 9: The conclusion requires a comprehensive rewrite to adequately convey the broader implications of the findings. The authors should consider suggesting practical applications of their research and clearly delineate potential future research directions, enriching the overall impact of the study.
Response 9: We have revised the conclusion to more comprehensively to highlight the broader implications of our findings, including potential practical applications in cancer immunotherapy. Additionally, we have outlined future research direction such as long-term efficacy studies, expanded safety evaluations, and translational development plans to strengthen the impact and relevance of the study.

Round 2
Reviewer 2 Report
Comments and Suggestions for Authors
Thank you for the revision. Perhaps it can be considered. I leave the final decision to the editor
Reviewer 4 Report
Comments and Suggestions for Authors
The authors made significant revisions based on the reviewers' comments; the manuscript can be accepted for publication.